# Asynchrony and functional diversity couple herbivore community dynamics to host plant diversity

Ming-Qiang Wang [1,2,3], Georg Albert [3], Carlo L. Seifert [3], Douglas Chesters [2], Helge Bruelheide [4,5], Yi Li [6], Jing-Ting Chen[2,7], Andréa Davrinche [4,5,8], Sylvia Haider [9], Shan Li [6], Goddert von Oheimb [10], Tobias Proß[4], Keping Ma [6], Xiaojuan Liu [6], Arong Luo [2], Andreas Schuldt [3] ✉ & Chao-Dong Zhu [2,11,12] ✉

Biodiversity loss can destabilize ecosystem functioning. How biodiversity–stability relationships are interlinked across trophic levels remains poorly investigated, however, limiting our ability to predict ecosystem-level consequences of declining biodiversity. Here, we analyze the drivers of multi-year herbivore community stability—as a key connector between primary producers and higher trophic levels—and its coupling with host tree diversity and growth stability along a subtropical tree diversity gradient. Phylogenetic diversity, abundance asynchrony and population stability of herbivores emerge as key intra-community regulators of herbivore temporal stability. These regulators, in turn, are strongly affected by changes in tree species richness through tree functional diversity, tree growth asynchrony, and tree growth population stability. Importantly, accounting for herbivore dietary specialization unveils clear stabilizing effects of tree species richness on the community stability of specialists but not of generalists. For the overall herbivore community, higher tree richness results in less stable abundance dynamics. Our findings suggest that biodiversity loss will propagate bottom-up to affect the stability of communities at higher trophic levels, and particularly destabilize communities of more vulnerable specialists. Global change and plantation management may thus also compromise biodiversity conservation by reducing abundance and species richness stability of higher trophic levels.

In times of global change—characterized by an increased severity of biodiversity loss, natural disturbances, and pest outbreaks[1–3]—maintaining the stability of ecosystem functioning has emerged as a key environmental and societal challenge[4]. Early theory[5] and recent empirical research[6,7] have emphasized the central role that biodiversity can play in promoting ecosystem stability. Such stabilizing effects may operate through mechanisms such as averaging effects, negative covariance among species, and insurance effects[8] and suggest that the current biodiversity crisis[9,10] might have long-term negative consequences for ecosystem stability and ultimately human well-being[6,11]. The full consequences, however, are difficult to assess. This is because, for one, biodiversity–stability relationships may vary with environmental (e.g., climate) conditions and across spatial and temporal scales[7,12,13], and we currently lack insights from many ecosystems.

Moreover, most empirical studies on diversity–stability relationships have focused on individual trophic levels, mostly the primary producer level[12,14]. We still lack a thorough knowledge on the extent to which diversity–stability relationships across trophic levels are interlinked through food webs[15]. Since trophic interactions are key drivers of ecosystem functions[16], such insights are essential to develop a clear understanding of how biodiversity affects ecosystem stability.

For ecosystem functioning and stability, the interactions between plants and herbivores are of particular relevance as the latter indirectly link producers with higher trophic levels[17,18]. The temporal stability (i.e., invariability in time at community and population level) of herbivore abundance can substantially alter ecosystems via stability of herbivory[19,20]. This, for instance, can be well-illustrated by forests where recent climate change effects have led to herbivore outbreaks that, in turn, caused large-scale forest diebacks[3]. The temporal stability of herbivore communities is also important for the provisioning of food for higher trophic levels and the long-term maintenance of biodiversity[21]. Herbivore temporal stability is often low in monocultures, whereas herbivore communities are assumed to be more stable in plant mixtures[15]. Monoculture plantations represent a dominant forest management practice worldwide[22], but their low biodiversity may reduce their capacity to buffer against environmental fluctuations and pest outbreaks. The regulatory mechanisms driving the dynamics of trophic interactions between plants and herbivores across temporal scales are, however, not well explored at the community level[23].

For tree communities, the temporal stability in growth and productivity has recently received increasing attention[14,24]. In general, community stability can be partitioned into species asynchrony, i.e., a temporal misalignment of species dynamics, and the stability of species' populations themselves[14]. It was shown that tree diversity stabilizes biomass productivity via compensatory effects of tree growth asynchrony (i.e., variability in the extent of temporal growth consistency among tree species) among functionally distinctive species and, to a lesser extent, population growth stability (i.e., the extent of temporal growth consistency within tree species)[14]. By contrast, the importance of such metrics in herbivore communities for determining stability of abundance and species richness over time remain largely unexplored for forest ecosystems. Even more so, the extent to which herbivore community dynamics are moderated by the diversity and temporal stability in the growth rates of their host plant communities remains unclear. Herbivore community stability might show a tight coupling with community asynchrony and population stability in plant growth as moderators of resource availability. It could be expected that generalist herbivores, which are less restricted in their host use, benefit from growth asynchrony of host species by stabilizing in both abundance and species richness over time[25]. Stability of dietary specialists, by contrast, might be positively affected by the population stability of their host plants[15]. However, empirical insights on how diversity and stability are interlinked between herbivores and their host plants are available primarily for grasslands, which are characterized by a strong yearly biomass turnover of herbs and graminoids[15]. As long-lived organisms, trees are rather constant and predictable resources, and associated herbivores might thus be less affected by changes in the stability of host biomass production. Composition and functional diversity of tree-feeding herbivores might therefore be more directly connected to host plant diversity, determining the availability of different resources[26]. This pattern could also be captured by phylogenetic relatedness among herbivore species, which can reflect herbivore functional dissimilarity and coevolutionary processes[27].

Here, we analyze the mechanisms underlying herbivore community stability in a species-rich subtropical forest ecosystem, assessing key properties of herbivore communities and the extent to which these are coupled to the diversity and growth stability of their host tree communities. We use time series data (17 replicated sampling seasons spread across six years from 2017 to 2022) of a dominant group of insect herbivores (lepidopteran larvae) and their host tree communities in a large-scale tree biodiversity experiment (BEF-China[28]). The experiment was conducted at two sites located 4 km apart, and our study included 52 randomly distributed plots reflecting a tree species richness gradient from monocultures to mixtures with up to 24 species. Specifically, we use linear models and path analysis to test the pathways that directly and indirectly (via tree and herbivore community asynchrony and population stability and via herbivore phylogenetic diversity) connect tree species richness and its effects on tree growth to the temporal stability of herbivore abundance and species richness (Fig. 1). Because we are interested in the relative role of direct and indirect effects of tree species richness, we test four alternative path models of decreasing complexity that sequentially dropped direct effects of tree species richness and growth on herbivore stability (Fig. 1c). We hypothesize that (1) the community dynamics of herbivores are coupled to the diversity and growth rate stability of their host tree communities, with metrics related to tree growth asynchrony and population stability explaining the stability of the associated herbivore community (Fig. 1a, b). As tree growth stability, in turn, depends on tree diversity, we further expect that (2) the loss of tree diversity indirectly (via tree growth) destabilizes herbivore communities (Fig. 1c). Finally, we expect that at the level of herbivore communities, (3) community stability is primarily driven by asynchrony among dominant generalist herbivores, with specialist species being much rarer and their stability depending more strongly on population stability.

## Results
In total, 17,850 caterpillars were collected over the sampling period of the 6 years. A total of 243 molecular operational taxonomic units (MOTUs) were included in our final analysis after excluding species represented by fewer than five individuals. Of these 243 MOTUs, 140 were classified as generalist herbivores (accounting for 9063 individuals) and 103 as specialists (with 4943 individuals).

### Tree diversity indirectly stabilizes herbivore communities
Linear models and subsequent path analysis showed that tree species richness ultimately moderated the community dynamics of herbivores. For the path models, the model variant excluding direct effects of all tree variables on herbivore stability metrics and direct effects of tree species richness on herbivore phylogenetic diversity (measured as mean pairwise phylogenetic distance, MPD) and population stability (model variant 4, Fig. 1c) received the highest support (Table S1). Consequently, the effects of tree species richness were largely indirect by influencing herbivore asynchrony, population stability, and herbivore phylogenetic diversity (or, alternatively, abundance/richness, see supplementary methods and sensitivity analysis results in Figs S1 and S2 and Tables S2–S7) via tree functional diversity as well as asynchrony and population stability of tree growth (Figs. 2, 3 and S3; Tables S8–S10).

### Divergent effects of tree diversity on generalists and specialists
Tree species richness was positively related to herbivore abundance and richness stability, particularly for specialist herbivores (Figs. 2c, 3c and 4). These positive effects were mediated by the strong link between tree species richness and tree growth asynchrony, and the positive effects of the latter on herbivore phylogenetic diversity and asynchrony, which in turn promoted herbivore stability. By contrast, the effects of tree species richness on generalist and, subsequently, also on overall herbivore stability were altogether non-significant, because positive effects via tree growth asynchrony (and tree growth population stability) were weaker and counteracted by negative effects via tree functional diversity (Fig. 4) on herbivore population

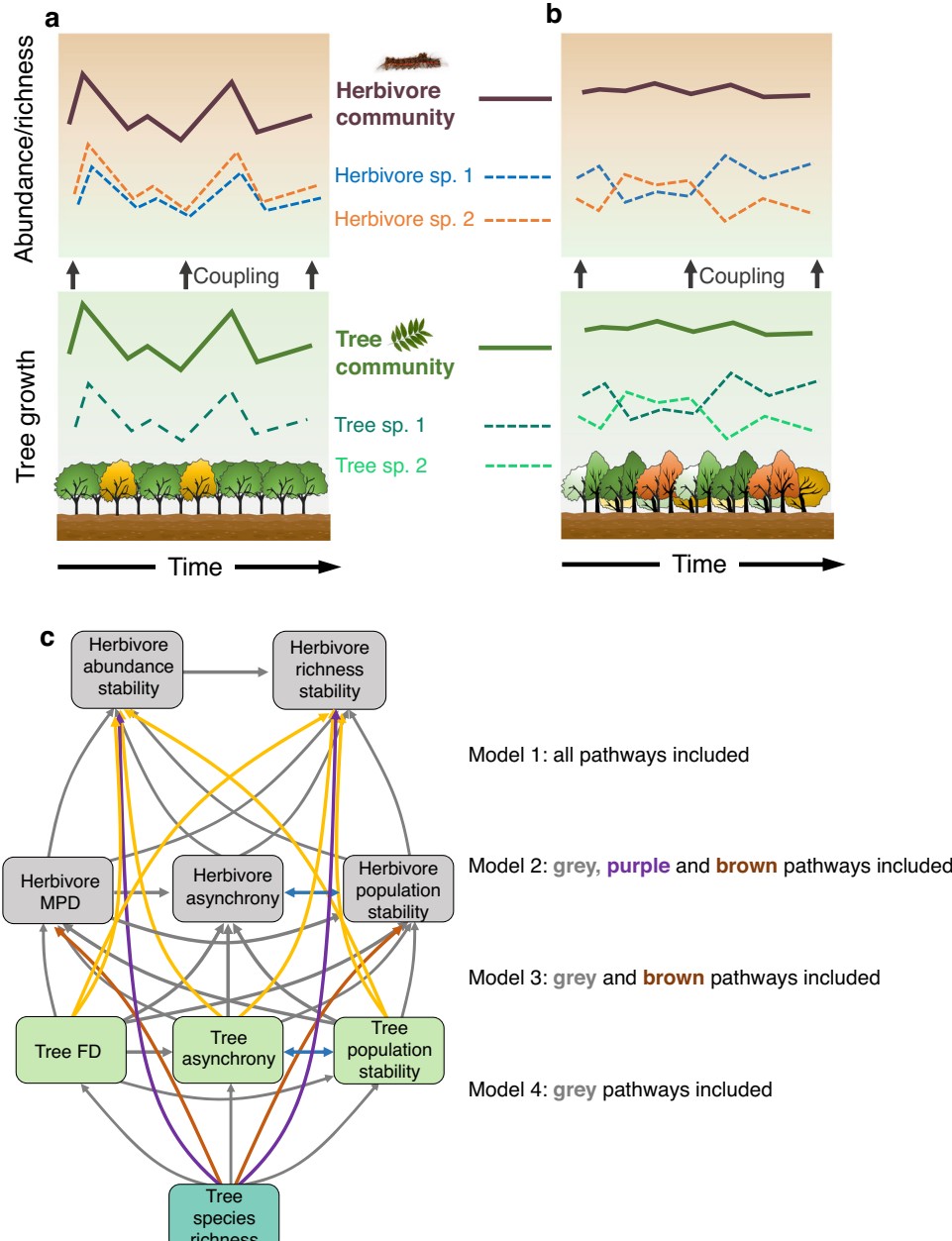

**Fig. 1 | Conceptual figure and initial path model structure illustrating the determining mechanisms of herbivore community stability.** Herbivore abundance asynchrony among species and community stability are coupled with the diversity and temporal growth rate stability of their host tree communities (Hypothesis 1), with herbivore community stability ultimately **a** destabilized by low tree species richness and **b** stabilized by high tree species richness (Hypothesis 2). Specifically, in **a** monocultures and species-poor mixtures, the pronounced fluctuations in tree growth may lead to reduced asynchrony among herbivore species, thereby destabilizing the herbivore community. Conversely, in **b** more diverse mixtures, complementary dynamics resulting from the asynchrony among tree species could contribute to the stabilization of plant communities. This stabilization effect extends to herbivore communities, promoting greater asynchrony among herbivore species and ultimately enhancing the stability of the herbivore community (Hypothesis 3). These relationships were investigated with **c** path models. Structure based on theoretical expectations and correlations among herbivore- and tree-based variables: mean phylogenetic diversity (MPD), species asynchrony, population stability of herbivores; tree functional diversity (FD), species asynchrony, population stability of trees. Arrows indicate expected causal relationships. Blue lines are covariances retained in the path models. We assessed with four alternative models how direct vs. indirect effects of host tree-based metrics influence herbivore community dynamics. Model 1: both direct and indirect pathways from trees to herbivore stability; Model 2: restricting tree growth effects to indirect pathways via herbivore population stability and asynchrony, except for tree species richness; Model 3: assuming even species richness acts only indirectly; Model 4: assuming that tree diversity influences herbivores solely through effects on tree functional diversity, asynchrony, and population stability. Model 4 was selected for our final analyses.

stability (Fig. 2b) and herbivore asynchrony (Fig. 2a). Interestingly, tree functional diversity was negatively related to herbivore asynchrony—with negative effects on overall and specialist patterns (Figs. 3a, c and Tables S8 and S10) and no effects on generalists (Fig. 3b and Table S9)—while tree asynchrony consistently promoted herbivore

phylogenetic diversity (Fig. 3 and Tables S8–S10). Average population stability of tree growth was negatively related to population stability of generalist and overall herbivores, whereas there was no relationship for specialists (Fig. 3 and Tables S8–S10). These patterns were consistent across additional sensitivity analyses (see supplementary

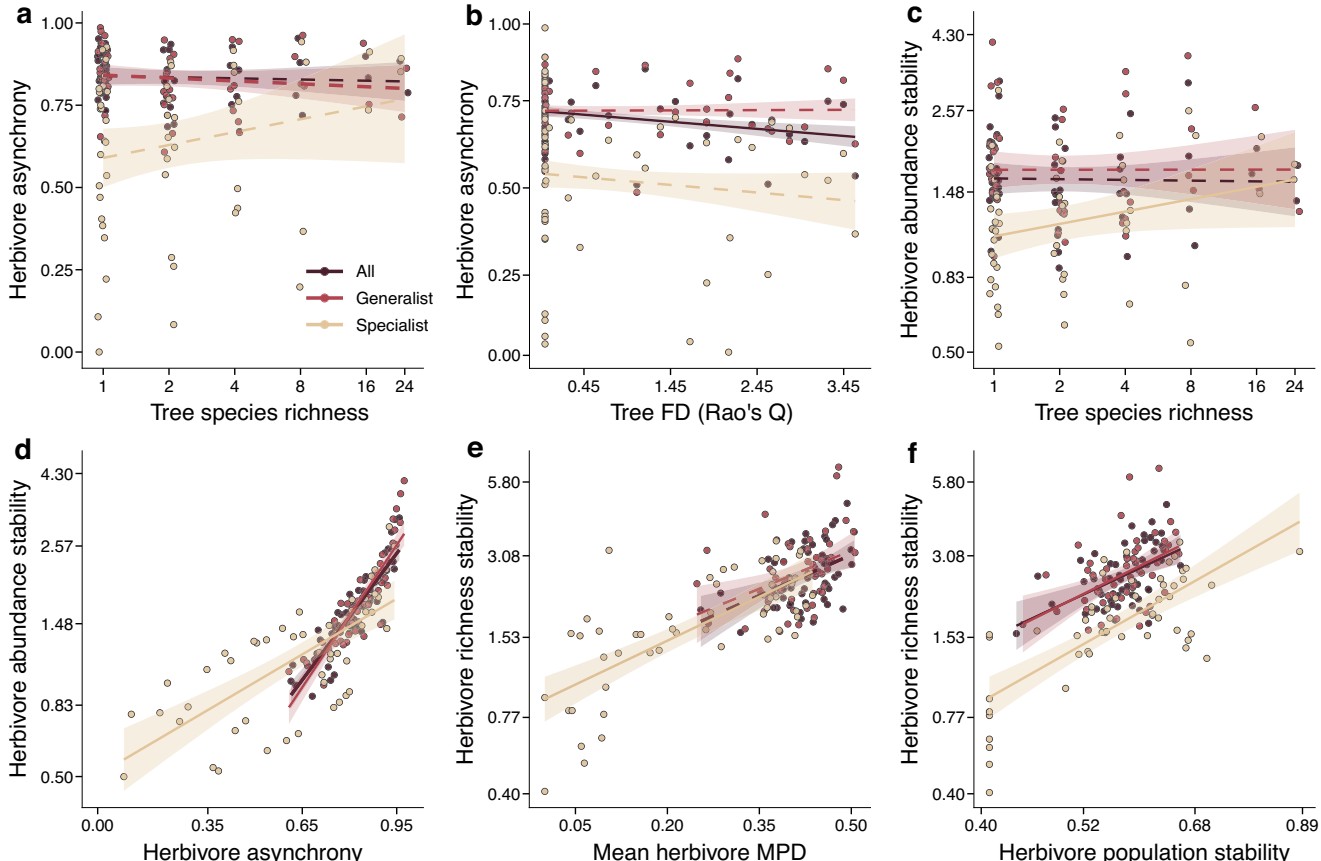

**Fig. 2 | Bivariate relationships between herbivore stability, asynchrony, population stability, mean phylogenetic diversity (MPD) and tree species diversity based on linear model results.** "All" includes the entire herbivore community (generalists and specialists) analyzed together (All: light brown, generalist: red, specialist: dark red). Lines are linear regression model fits of the relationships between herbivore asynchrony and **a** tree species richness and **b** tree functional diversity. Relationships between herbivore abundance stability and **c** tree species richness and **d** herbivore asynchrony. Relationships between herbivore richness stability and **e** herbivore MPD and **f** herbivore population stability. Regression lines (with 95% confidence bands) show significant (solid lines, $p \leq 0.05$) or non-significant (dashed lines, $p > 0.05$) relationships. Note that tree species richness, population stability, abundance stability and richness stability of herbivores were log-transformed. Error bands represent 95% confidence intervals around the mean.

method and note text), including the use of an alternative stability index, the inclusion of all caterpillar individuals in path models, and path models excluding monocultures.

### Herbivore asynchrony and phylogenetic diversity stabilize herbivore communities

Within the herbivore communities, abundance stability was particularly promoted by herbivore asynchrony (Fig. 2d and 3 and Table S11) and, to a lesser extent, by population stability in all cases (Figs. 3 and S3a and Table S11). Herbivore richness stability was strongly related to herbivore abundance stability in all cases (Fig. 3) and furthermore affected by both herbivore phylogenetic diversity (for overall herbivores and specialists; Figs. 2e and 3 and Table S12) and herbivore population stability (including for generalists; Figs. 2f and 3). Herbivore phylogenetic diversity positively affected herbivore population stability of specialist and overall herbivores (Figs. 3 and S3b and Table S13), and increased herbivore asynchrony for specialists (Figs. S2c and 3c and Table S14).

## Discussion

Our study unravels how the temporal stability of abundance and species richness of herbivore communities is linked to changes in the species richness of their host tree communities. Specifically, the results show that indirect effects of tree species richness via tree functional diversity and tree growth on herbivore phylogenetic

diversity, abundance asynchrony, and population stability play a prominent role in explaining these relationships. Importantly, our study highlights that the degree of dietary specialization strongly affects the way in which individual community properties impact stability. In particular, we found a clear and ecologically intuitive stabilizing effect of tree species richness on the temporal abundance and richness stability of specialist herbivores, whereas such effects were absent for generalists. For the overall herbivore community, path analysis indicated that this altogether resulted in less stable abundance dynamics at higher tree species richness via tree functional diversity, even though the direct relationship between tree species richness and herbivore stability was not significant in our linear models. These findings underscore the crucial role of host diversity in buffering specialist herbivores against environmental fluctuations and emphasize that biodiversity–stability relationships can vary markedly across trophic groups depending on their level of dietary specialization. Our study thus highlights the consequences that biodiversity loss at lower trophic levels can have for the community stability at higher trophic levels.

### Regulation of herbivores via host diversity and growth rate stability

Recent studies on temporal community dynamics have started to provide insight into the mechanisms that underlie the maintenance of stability of ecological communities, but largely focused on individual

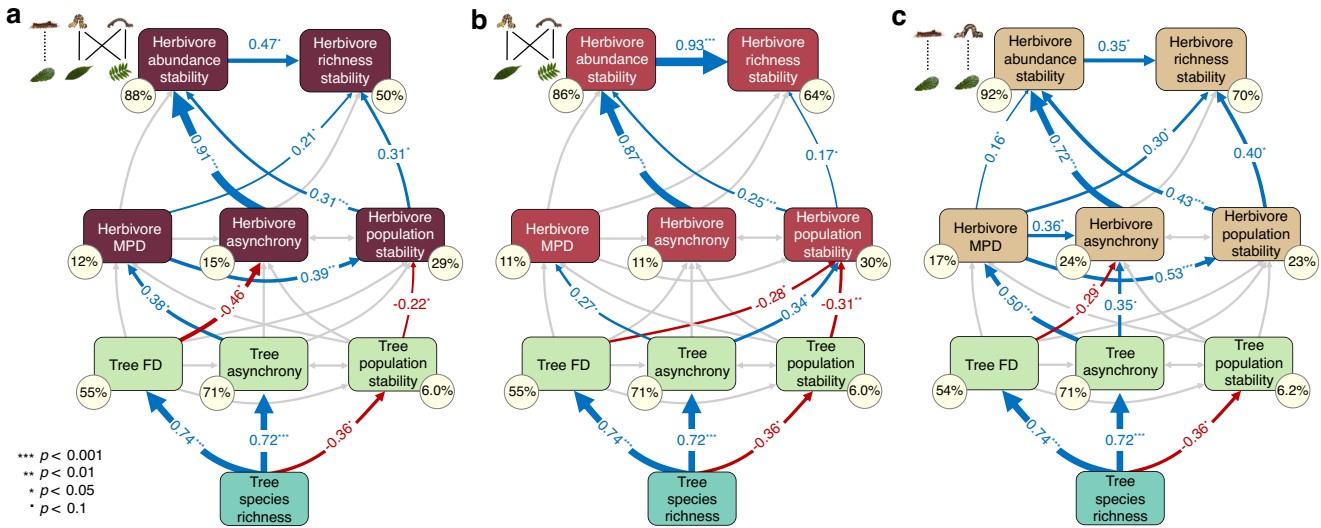

**Fig. 3 | Effects of tree diversity on herbivore community stability via bottom-up regulation based on path model results.** Potential effects of tree species richness (green rectangle), tree functional diversity (FD), species asynchrony and population stability (light green rectangles) on community stability of herbivore abundance and richness through mean herbivore abundance weighted phylogenetic diversity (MPD), species asynchrony and population stability for **a** overall herbivores ($\chi^2$ = 8.92, DF = 11, $P$ = 0.629; dark red rectangles), **b** generalist herbivores ($\chi^2$ = 12.42, DF = 11, $P$ = 0.333; red rectangles), and **c** specialist herbivores ($\chi^2$ = 8.53, DF = 11, $P$ = 0.507; light brown rectangles) based on path model results (see Tables S8–S10 for full results). Blue arrows indicate positive effects, red arrows show negative effects ($p \leq 0.1$), grey arrows show non-significant pathways ($p > 0.1$). Arrow width was scaled by the standardized path coefficients. The proportion of variance ($R^2$) is shown in yellow circles. Note that tree species richness, population stability, abundance stability and richness stability of herbivores were log-transformed. Significance levels: $p < 0.1$ ('), $p < 0.05$ ('), $p < 0.01$ (''), $p < 0.001$ ('''). Note that direct effects of tree variables on herbivore stability metrics were not supported in this model (model variant 4, Fig. 1c). Statistical tests were two-sided, and no adjustments were made for multiple comparisons.

trophic levels, such as plants or arthropod predators[14,29–31]. Our study confirms the key roles of abundance asynchrony and population stability for community stability, and an important structuring effect of functionally diverse communities, but goes beyond previous studies by uncovering important linkages across trophic levels. Tree growth asynchrony could stabilize herbivore communities by buffering resource availability at the population level, reducing competition within communities[32]. Moreover, our study highlights the added value of expanding biodiversity and ecosystem functioning relationships by explicitly considering stability relationships. While high average biomass production may reflect favorable conditions in some years, it may mask substantial interannual fluctuations in growth[33]. The repercussions of tree diversity on herbivore diversity and stability we observed can thus have important ecological consequences—for example, with respect to predicting the probability of pest outbreaks following climate extreme events and subsequent tree growth variation[34]—that would not be evident when ignoring such a temporal perspective. Understanding what promotes stability allows us to assess how ecosystems buffer disturbances, maintain trophic interactions, and preserve long-term biodiversity under changing environmental conditions.

We found clear signals of the hypothesized (H1) bottom-up regulated coupling between the dynamics of herbivore communities and the diversity and growth rate stability of their host tree communities. Our study provides empirical evidence to confirm expectations that are so far largely based on ecological theory[35,36]. Specifically, our study highlights how interactions across trophic levels, as well as biodiversity loss at lower trophic levels, can propagate to affect temporal community dynamics at higher trophic levels. We largely observed indirect linkages between herbivore community metrics and host tree species richness. These linkages were driven by effects of tree species richness on tree functional diversity and the temporal stability in the host trees' growth rates, which in turn influenced herbivore phylogenetic diversity and abundance stability (asynchrony and population stability). Interestingly, tree functional diversity

particularly affected herbivore asynchrony and population stability, whereas tree growth asynchrony had strongest effects on herbivore phylogenetic diversity. These patterns indicate that herbivore community metrics can be only partially predicted from the corresponding metrics of their host trees.

Tree growth asynchrony promoted herbivore phylogenetic diversity of both generalists and specialists, suggesting that temporal stability in the availability of food resources might provide more niche opportunities that favor the coexistence of more distantly related herbivore species and, in consequence, functionally more diversified consumer communities with more distant phylogenetic relationships[37]. In the sensitivity analyses replacing herbivore phylogenetic diversity by herbivore abundance or richness (Figs. S1 and S2), the driving role of tree species richness behind these effects became even more evident. This was possibly due to the fact that these simpler metrics of biodiversity were not able to resolve the indirect pathways via tree asynchrony as accurately as a biodiversity metric accounting for phylogenetic (and functional) differentiation among herbivore species[38,39], underlining the importance of considering multiple metrics of biodiversity. Notwithstanding, the strong influence of tree species richness emphasizes the consequences that biodiversity loss at the host level can have for higher trophic-level diversity and, ultimately, stability.

The clearer, negative effects of tree functional diversity on overall herbivores and specialist asynchrony compared to generalist asynchrony may reflect limitations in the ability, particularly of specialist herbivores, to exploit multiple tree species effectively. The lack of clear effects of tree metrics on the abundance asynchrony of generalist herbivores suggests that the wider food spectrum of these species reduces the dependence on their hosts' asynchronous population dynamics. One reason could be that this wider food spectrum reduces competition among co-occurring herbivores[40]. Interestingly, the population stability of generalist herbivores strongly responded to tree functional diversity and tree growth stability. The negative effects of tree functional diversity indicate that lepidopteran generalist

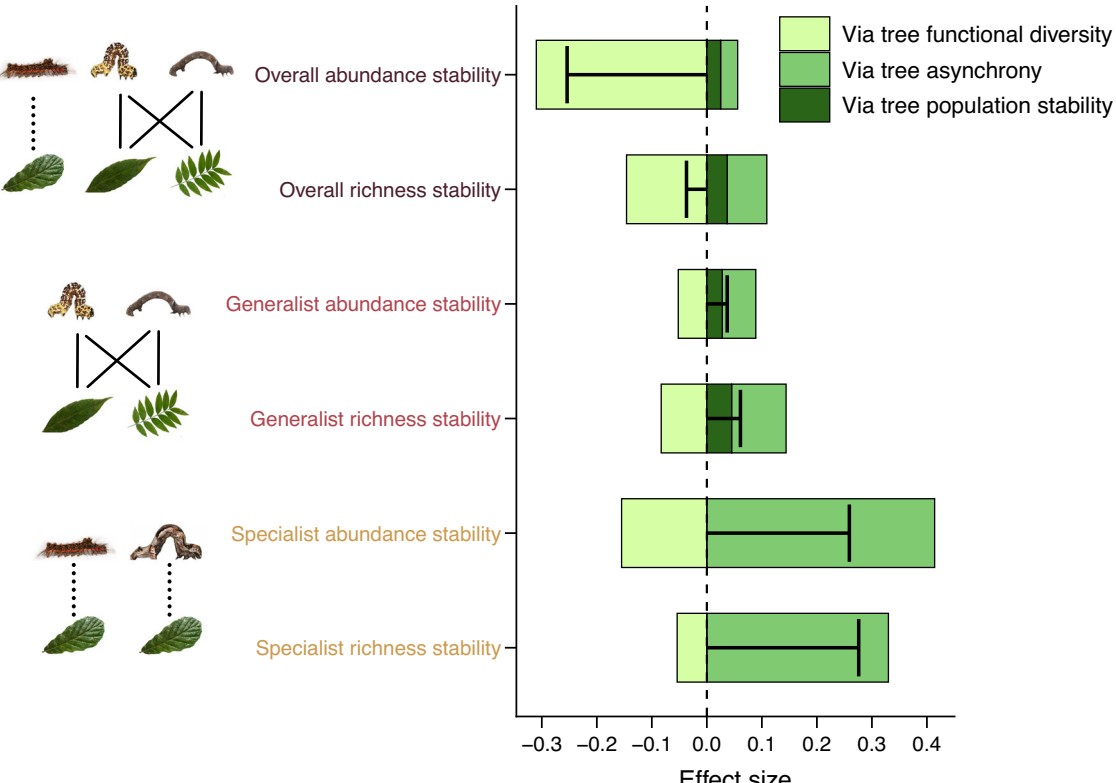

**Fig. 4 | Effects of tree species richness on herbivore richness stability from path models.** Bars show summed effects of tree species richness on the abundance and richness stability of all, generalist, and specialist herbivores, respectively. Effect sizes were calculated by summing indirect effects of tree species richness via tree functional diversity, tree asynchrony, tree population stability, herbivore mean phylogenetic diversity, herbivore asynchrony, herbivore population stability, and herbivore abundance stability. The different colors show effects of tree species on herbivore stability via tree functional diversity (light green), tree asynchrony (green), and tree population stability (dark green), respectively. Effect sizes were calculated as the product of standardized path coefficients connecting each predictor with herbivore components, summed over the individual predictors of each component for positive and negative effects on herbivore stability metrics, respectively. Black T-shaped lines indicate the total effects of tree species richness on herbivore stability metrics. Note that tree species richness, population stability, abundance stability, and richness stability of herbivores were log-transformed. See Supplementary note and Fig. S9 for effects including non-significant pathways. Source data are provided as a Source data file.

herbivore species, despite feeding on multiple tree species, favor certain functional types of trees[41]. Simultaneously, generalist herbivore species seem to benefit from switching between host tree species[42], as generalist herbivores had more stable populations when tree growth varied inter- and intra-specifically (i.e., asynchrony and population stability). This would also reduce competition among herbivore species[43]. Therefore, the consistent negative effects of tree functional diversity on both overall and generalist herbivore stability suggest that increased functional differentiation among trees creates greater heterogeneity in resource traits, which in turn destabilizes herbivore populations when they depend on specific host functional types. Similar findings have been reported in other multitrophic studies, where high functional diversity of basal resources increased variability in consumer communities[13,15]. Importantly, while tree species richness had a stabilizing effect on specialist herbivores, the net effect of tree diversity on overall herbivore stability was weak or even negative, highlighting that positive biodiversity–stability relationships may not uniformly extend across all consumer guilds. Overall, our findings indicate that there are strong bottom-up regulatory dynamics at play that are induced by tree species richness and potentially influence herbivore communities by modulating the levels of competition among them.

The resulting overall direct and indirect effects via tree growth of tree species richness on specialist herbivore community stability were clearly positive, partly supporting our expectation (H2) and underlining the crucial role that tree species richness may play in preventing outbreaks of often specialized insect pests[44]. The overall effects of tree species richness on generalists and thereby on overall herbivore community stability were less clear-cut due to counteracting influences via tree functional diversity and tree growth asynchrony, and they were even negative in the case of overall abundance stability. Realized tree species richness effects on herbivore community stability may therefore largely depend on the ratio of specialists to generalists. In addition to the commonly positive effects of tree richness on fostering species diversity at adjacent trophic levels[45,46], tree species richness may therefore also contribute to stabilizing the temporal dynamics of higher trophic levels, indicating clear long-term benefits for conserving biodiversity.

**Intra-community regulation of herbivore temporal dynamics**
Asynchrony and population stability also emerged as key drivers of community stability within the herbivore trophic level. Asynchrony was the main driver of abundance stability for both generalists and specialists, although we expected (H3) population stability to be particularly relevant for the latter. This deviation may be explained by the fact that many of our study plots were mixtures of different tree species, which hosted distinct herbivore assemblages. In such mixed-species plots, mean plot-level herbivore abundance (calculated across all herbivore species) does not necessarily rely on stable populations of each species. Instead, it can be maintained when fluctuations in abundance are asynchronous among species—even among specialists —because multiple host tree species support different herbivore

populations. These compensatory dynamics buffer total herbivore abundance over time, reducing the apparent influence of population stability on plot-level stability[25]. Our sensitivity analysis confirms this assumption, showing that effects of tree population stability were reduced even more when excluding monocultures (for which tree growth asynchrony equates to zero, as there cannot be variation among multiple tree species).

In line with our hypothesis (H3), herbivore phylogenetic diversity, in turn, promoted both asynchrony and population stability within the herbivore communities, which was most obvious for specialist herbivores. Phylogenetic diversity can be assumed to reflect functional differentiation[47]. Such niche differentiation can foster asynchronous population dynamics of species that are more different in their resource use[29]. At the same time, niche differentiation can reduce competition and thus stabilize the population sizes of abundant species[48]. This also means that losing phylogenetic or functional diversity of herbivores, for example as a consequence of human alterations of ecosystems and subsequent biotic homogenization[49,50], may destabilize herbivore community dynamics and increase the risk of pest outbreaks, especially for specialist herbivores.

Our findings of a clear coupling between herbivore community dynamics and the species richness and growth rate stability of their host trees suggest that biodiversity loss at lower trophic levels might destabilize community dynamics of higher trophic levels, in our case, particularly those of specialists. Overall, herbivores and generalists were less affected than specialists, although we essentially found two opposing pathways (via host functional diversity and host asynchrony) that may turn out to affect overall stability differently if host species loss happens in non-random ways. At the same time, global environmental change and plantation-focused forest management strategies, which lead to tree diversity loss, may jeopardize biodiversity conservation by reducing species richness stability over time. Our findings suggest that monoculture plantations, despite their widespread use in forest management[22], may be more prone to herbivore instability driven by specialist herbivores that can rapidly build up on their preferred hosts under low tree diversity. In contrast, mixed-species plantings dilute host availability and thereby stabilize herbivore communities, highlighting the value of biodiversity-oriented management for more resilient forest ecosystems.

## Methods

### Experimental site and design

We collected data from the BEF-China tree diversity experiment[28], which is located at Xingangshan, Jiangxi Province, southeast China (29°08′–29°11′ N, 117°90′–117°93′ E). The experiment comprises two study sites, namely site A and site B, each spanning approximately 20 ha, and established in 2009 (site A) and 2010 (site B). The study sites are characterized by seasonal monsoon climate with a mean annual temperature of 16.7 °C and a mean annual precipitation of 1800 mm[51]. The region is characterized by highly diverse native subtropical forests, predominantly composed of a blend of broadleaved evergreen and deciduous tree species.

There are 566 study plots (25.8 × 25.8 m) established on the two sites, forming a tree species richness gradient ranging from monocultures to mixtures of up to 24 tree species. Within each plot, 400 saplings were planted, arranged in 20 rows and 20 columns with a mean planting distance of 1.29 m. Altogether, 40 locally common tree species were planted across the two study sites. Tree species compositions at the two sites were largely non-overlapping, with different species pools of 16 broadleaved species at each site (and an overlap of eight tree species in the 24-species mixtures that were planted on both sites). In order to achieve a comprehensive representation of species across diverse levels of biodiversity, compositions of mixtures were randomly assigned based on a "broken-stick" design[28]. The allocation of tree species to planting positions within each plot was done

randomly, and the total number of individuals per plot was evenly distributed among the planted species.

For our study, we selected a subset of 64 plots (32 per site) which followed a broken-stick design and spanned the tree diversity gradient from monocultures to 24 species-mixtures. Per site, we considered sixteen monocultures, as well as eight 2-, four 4-, two 8-, one 16-, and one 24-species mixtures. We excluded twelve plots due to high tree mortality, resulting in a final set of 52 plots included for our analyses.

### Sampling, species delimitation and phylogenetic analyses

We collected lepidopteran larvae, a key group of insect herbivores in forest ecosystems, across 17 sampling events spread across six years (from 2017 to 2022). To capture seasonal changes in caterpillar assemblages and gain a rather comprehensive picture of the occurring fauna, we sampled three times per year (spring: April, summer: June, and autumn: September; April in 2020 was missed because of the COVD-19 pandemic). Caterpillars were sampled by beating individual tree branches with a padded stick and subsequent collecting all dislodged individuals that fell on a white sheet (1.5 × 1.5 m), placed directly beneath the branches[52]. To standardize the sampling effort, we started collecting the trees in the first row of each sampled plot and continued until 80 living trees were sampled. Because of the random planting design allowed us to adequately cover the complete tree species composition per plot. Collected caterpillars were individually preserved in tubes filled with 99.5% ethanol and stored at −20 °C until further processing.

We sequenced the barcode region (i.e., mitochondrial cytochrome c oxidase subunit I (COI), 658 bp) of all caterpillars using universal primer pairs, LCO1490 and HCO2198[53]. Polymerase chain reactions (PCR) were conducted in 96-well plates with a total volume of 30 µl, consisting of 10 µl ddH₂O, 15 µl Premix PrimeSTAR HS (TaKaRa), 1 µl of each primer (10 µM), and 3 µl of genomic DNA template. The thermocycling conditions were as follows: an initial denaturation at 94 °C for 2 min; 29 cycles of 94 °C for 50 s, 50 °C for 50 s, and 72 °C for 1 min; and a final extension at 72 °C for 6 min. All reactions were run on an Eppendorf Mastercycler Gradient. PCR products were examined on a 1% agarose gel, and samples showing clear single bands were purified and sequenced using BigDye v3.1 chemistry on an ABI 3730xl DNA Analyzer (Applied Biosystems). Based on the obtained sequences, caterpillars were then assigned to molecular operational taxonomic units (MOTUs), which are widely used in ecological studies and have been proven useful for taxa or regions where taxonomic knowledge is largely incomplete[54]. We used three methods to generate MOTUs and then chose the most consistent clustering method for further analysis, they are: threshold-based hierarchical clustering with BLASTclust, Automatic Barcode Gap Discovery (ABGD), and Poisson Tree Processes model (PTP). In total, 885 distinct MOTUs were identified using the most consistent method of molecular species delimitation.

As a proxy for herbivore functional diversity, phylogenetic diversity is a useful surrogate[27,55]. To gain more accurate estimates of phylogenetic diversity, we first used RaxML[56] to place our MOTUs onto a comprehensive (11,000 species) synthesis phylogeny of Lepidoptera, which was obtained from https://insectphylo.org/ (Chesters et al.[57]). We then calculated abundance weighted herbivore phylogenetic diversity (i.e., mean pairwise phylogenetic distance; MPD) for sets of the phylogenetically-placed herbivore MOTUs at community level.

### Host specialization estimation

To evaluate host specificity of individual caterpillar species, we calculated the wood volume weighted MPD of host trees utilized by a given caterpillar species. MPD-based indices are commonly employed to quantify host plant specialization of insect herbivores (e.g.[58–61]). The weighted host MPD considers both frequency and phylogenetic relatedness of the host plants. The highest level of dietary

specialization is indicated by host MPD values of zero, whereas the degree of dietary generality increases with increasing host MPD. Weighted host MPD values were calculated for every caterpillar species, using the *'mpd'* function as implemented in the picante package[62]. To do so, we used a maximum likelihood phylogeny covering all caterpillar species included in this study. For each caterpillar species, host plant records across all plots were considered to calculate its degree of dietary specialization.

Before classifying species as either specialists or generalists, we first excluded species with fewer than five individuals. This way, we aimed at eliminating the bias caused by rare species (i.e., that generalists with low abundances are erroneously considered as specialists). Subsequently, we applied hierarchical clustering based on host MPD using partitioning around medoids (PAM)[63]. This method uses algorithms to group taxa into well-defined 'hard' clusters with minimal prior input, thereby enhancing the objectivity of the classification procedure. Clustering analyses were carried out using two clusters ($k = 2$, i.e., generalists and specialists). The threshold value was then used to separate specialists (host MPD < 0.181, 103 species) from generalists (host MPD > 0.181, 140 species).

There are other and often more complex phylogenetic indices that could be used to reflect herbivore diet breadth (e.g.,[64]), but we chose the classification as described above to allow for an easier comparison with earlier studies that commonly applied a binary demarcation for dietary breadth as well (e.g.,[65]).

### Tree growth and tree functional diversity
The aboveground wood volume ($V_i$) of an individual tree $i$ was measured annually between 2016 and 2021. It was determined by the formula:

$$V_i = H_i \times \pi (BR_i)^2 \qquad (1)$$

where $H_i$ denotes the height of the tree, and $BR_i$ represents its basal radius measured at ground level, with both measurements in meters. For each designated plot and corresponding year, the total tree volume, denoted as $V$ (m³/ha), was determined by summing the wood volumes of the living trees ($V_i$) in the central 36 planting positions of the plot. This calculation was standardized based on the plot area.

The absolute growth rates ($TGR_i$) of individual trees were calculated for both monocultures and mixtures within the plot:

$$TGR_i = V_i, t_2 - V_i, t_1 \qquad (2)$$

where $V_i t_1$, and $V_i t_2$ are the tree wood volumes at two consecutive years, $t_1$ and $t_2$, within the sampling period from 2016 to 2021. $TGR_i$ was used for the calculation of tree asynchrony and population stability.

We used six functional traits for tree functional diversity calculation, namely specific leaf area (SLA), leaf dry matter content (LDMC), ratio of leaf carbon and nitrogen content (C:N), leaf magnesium content (Mg), leaf tannin content, and leaf lignin content. These functional traits have been recognized to reflect host plant palatability and are often closely related to herbivore performance and degree of herbivory[66,67]. We measured the functional traits at three to five different tree heights per species in 2018[68]. Tree functional diversity was calculated on plot level as the mean pairwise distance between mean trait values per tree species, weighted by tree wood volume, and expressed as Rao's Q[69] (note that Rao's Q is 0 in monocultures).

### Temporal stability, population stability, and asynchrony calculation
We calculated the temporal stability of herbivores based on the overall abundance and richness of herbivores at plot level and per year (i.e., multiple measures per year were pooled). The temporal stability was

calculated as the inverse of the coefficient of variation

$$\text{stability} = \mu / \sigma \qquad (3)$$

where μ and σ are the mean value and standard deviation, respectively, of herbivore abundance and herbivore richness at plot level and across sampling years. The metrics should thus be understood as stability of herbivore communities, where higher values denote a more stable community.

The plot level-average population stability of tree growth and herbivores was calculated as the sum of all species' temporal stability indices (applying formula 3).

We calculated species asynchrony (hereafter asynchrony) at community level for both trees and herbivores using the species synchrony statistic φ[70] as 1 – φ

$$\text{asynchrony} = 1 - \frac{\sigma^2}{\left(\sum_{i=1}^{n} \sigma_i\right)^2} \qquad (4)$$

where σ is the overall community standard deviation across years and $\sigma_i$ is the temporal standard deviation of the annual absolute tree growth (for trees) or abundance (for herbivores) of species $i$ in a plot of $n$ species. Therefore, asynchrony will increase if the variance in individual tree species growth or herbivore abundance increase, corresponding to the variance in tree growth or herbivore abundance at community level. The asynchrony ranges from 0 to 1, representing a gradient from complete synchrony to complete asynchrony (note that asynchrony value is 0 in monocultures).

### Modeling and statistical analysis
We used linear models to check the effects of tree species richness, functional diversity, population stability, and herbivore phylogenetic diversity on herbivore asynchrony and population stability for all herbivores, generalist herbivores, and specialist herbivores, respectively. Moreover, we also included herbivore asynchrony and population stability as additional predictors and checked their effects on temporal stability of herbivore abundance and richness. We did not include tree asynchrony as a predictor in the linear models due to the strong correlation with tree species richness (Pearson's $r = 0.77$, $P < 0.001$). We included the interactions between site and tree species richness, site and tree functional diversity, site and tree population stability as further predictors. We streamlined the linear models through a stepwise process guided by the Akaike Information Criterion, corrected for small sample sizes (AICc). Subsequently, we selected subset models characterized by the lowest AICc values. In addition, we used single linear models to directly test the relationships between tree species richness, abundance, and richness stability of herbivores (separately for all herbivores, generalists, and specialists).

To improve normality and variance homogeneity of the model residuals, population stability, abundance stability, and richness stability as response variables were log-transformed. Likewise, tree species richness as a predictor was log-transformed in all analyses. To enable direct comparisons of model estimates, all continuous predictors were standardized (mean = 0, standard deviation = 1) prior to the analyses.

We used path analyses to disentangle direct and indirect drivers of the plant community (i.e., tree species richness, functional diversity, asynchrony, and population stability) on herbivore stability (abundance stability and richness stability) through herbivore phylogenetic diversity, asynchrony, and population stability. In the absence of herbivore trait measurements, this allowed us to test the potential effects of herbivore functional diversity on herbivore dynamics (keeping the potential links similar to trees).

We constructed our initial model according to the current knowledge on mechanisms driving biodiversity-stability relationships for

plants, insects, and ecosystem functions[14,29,48]. We assumed that tree species richness influences tree functional diversity, tree asynchrony and tree population stability (Fig. 1). Moreover, we assumed that all tree-based predictors can directly affect herbivore phylogenetic diversity, herbivore asynchrony, herbivore population stability and herbivore stability (Fig. 1). The herbivore phylogenetic diversity were separately included in the models to reveal the potential different roles of herbivore community metrics in community stability. For both trees and herbivores, we hypothesized that herbivore phylogenetic diversity can directly affect asynchrony and population stability. We considered covariances between asynchrony and population stability in all models.

To test for effects of feeding specialization on herbivore-host tree interactions, three separate models were fitted based on the initial model structure described above: one for all herbivores, one for generalists, and one for specialists. Moreover, to assess the level of support for different ways in which host tree-based metrics influence herbivore community dynamics, we compared four alternative model structures that successively removed interactions between different levels of host tree and herbivore community data (Fig. 1): First, we considered the initial model with all potential links included (Model 1). Second, we assumed that effects are entirely mediated by stability components, except for effects such as microclimate buffering that may remain in tree species richness. Therefore, we removed the direct connections between trees and herbivore abundance and richness stability, except for effects of tree species richness (Model 2). Third, we assumed that there are no such effects of tree species richness, so we additionally excluded the direct pathways between tree species richness and herbivore stability metrics (Model 3). Fourth, we assumed that tree species richness only has indirect effects via tree stability components and functional composition. We hence removed the links between tree species richness and herbivore phylogenetic diversity and population stability (Model 4). The same model structure selected for overall herbivores was applied to generalists and specialists to ensure comparability across groups. Model fit statistics were summarized (Tables S1 and S15). The best-fitting model was selected based on the lowest AIC value[71]. We used bootstrapped $P$ values based on 1000 bootstrap draws to ensure robust results[72]. The effect sizes of tree species richness on herbivore richness stability were then summarized. We additionally calculated effect sizes, including nonsignificant pathways, and assessed their variability using 1000 bootstrap draws (see supplementary for the results).

We ran additional sensitivity analyses for the path models in which we assessed the role of the herbivore diversity metric and the potential methodological influence of monoculture plots, respectively (see Supplementary Methods and note).

All analyses were conducted in R 4.2.2 (www.r-project.org) with the packages picante[62], vegan[73], FD[74], and lavaan[75].

## Reporting summary

Further information on research design is available in the Nature Portfolio Reporting Summary linked to this article.

## Data availability

The tree and herbivore data generated in this study have been deposited in the figshare[76] at https://doi.org/10.6084/m9.figshare.30531410, in the Science Data Bank[77] at https://doi.org/10.57760/sciencedb.31116, in the BEF-China repository at https://data.botanik.uni-halle.de/bef-china/datasets/695. The COI sequences generated in this study have been deposited in The Genome Sequence Archive (GSA) under project PRJCA052105 (accession ID: CRA034950). Source data are provided with this paper.

## Code availability

Code is available on the Science Data Bank[77] at https://doi.org/10.57760/sciencedb.31116.

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

## Acknowledgements

We thank field assistants (Yinquan Qi, Wanxi Wang, Zhangrong Cheng) for their helps in the sampling. We acknowledge tremendous help and the support of the BEF-China platform of the Zhejiang Qianjiangyuan Forest Biodiversity National Observation and Research Station. This study was supported by the National Key Research Development Program of China (2022YFF0802300), the Science & Technology Fundamental Resources Investigation Program of China (2023FY100203), the National Natural Science Foundation, China (32100343), the Key Program of the National Natural Science Foundation of China (No. 32330013) and the German Research Foundation DFG (SCHU 2609/4-1) within the MultiTroph research unit (452861007/FOR 5281). MQW was supported by the Alexander von Humboldt research fellowships. CDZ's lab has been continuously supported by grants from the Key Laboratory of the Zoological Systematics and Evolution of the Chinese Academy of Sciences (grant number 2008DP173354) and the Key Laboratory of Animal Biodiversity Conservation and Integrated Pest Management (Chinese Academy of Sciences, SKLA2501). AD and TP were supported by the International Research Training Group TreeDì (GRK2324) jointly funded by the Deutsche Forschungsgemeinschaft (DFG, German Research Foundation)—319936945 and the University of Chinese Academy of Science (UCAS). Some early data were sampled by supports from the Strategic Priority Research Program of the Chinese Academy of Sciences (XDB310304) and the National Science Fund for Distinguished Young Scholars (31625024).

## Author contributions

A.S., C.D.Z., and M.Q.W. conceived the idea for the manuscript; C.D.Z., A.S., and M.Q.W. designed the study. M.Q.W., H.B., J.T.C., A.D., S.H., S.L., Y.L., G.v.O., T.P., K.M., X.L., and A.L. collected and/or contributed data; D.C. and M.Q.W. conducted the phylogenetic analysis; M.Q.W. and C.L.S. estimated the host specialization; M.Q.W. conducted the statistical analyses and wrote the manuscript, with input from G.A., A.S., and all coauthors.

## Competing interests

The authors declare no competing interests.

## Additional information

[1]Mountain Ecological Restoration and Biodiversity Conservation Key Laboratory of Sichuan Province, Chengdu Institute of Biology, Chinese Academy of Sciences, Chengdu, China. [2]State Key Laboratory of Animal Biodiversity Conservation and Integrated Pest Management, Chinese Academy of Sciences, Beijing, China. [3]Department of Forest Nature Conservation, University of Göttingen, Göttingen, Germany. [4]Institute of Biology/Geobotany and Botanical Garden, Martin Luther University Halle-Wittenberg, Halle (Saale), Germany. [5]German Centre for Integrative Biodiversity Research (iDiv) Halle-Jena-Leipzig, Leipzig, Germany. [6]State Key Laboratory of Vegetation and Environmental Change, Institute of Botany, Chinese Academy of Sciences, Beijing, China. [7]College of Biological Sciences, University of Chinese Academy of Sciences, Beijing, China. [8]Research Center for Ecological Change, Organismal and Evolutionary Research Programme, Faculty of Biological and Environmental Sciences, University of Helsinki, Helsinki, Finland. [9]Institute of Ecology, Leuphana University of Lüneburg, Lüneburg, Germany. [10]Institute of General Ecology and Environmental Protection, TUD Dresden University of Technology, Tharandt, Germany. [11]State Key Laboratory of Integrated Pest Management, Institute of Zoology, Chinese Academy of Sciences, Beijing, China. [12]Key Laboratory of Animal Biodiversity Conservation and Integrated Pest Management, Chinese Academy of Sciences, Beijing, China.
✉e-mail: andreas.schuldt@forst.uni-goettingen.de; zhucd@ioz.ac.cn

