## [Transparent Peer Review file · Nature Communications]

Asynchrony and functional diversity couple herbivore community dynamics to host plant diversity

Corresponding Author: Dr Ming-Qiang Wang

Version 0:

Reviewer comments:

Reviewer #1

(Remarks to the Author)

This study investigates the multitrophic relationships linking tree species richness, functional diversity, tree temporal dynamics, herbivore functional diversity, herbivore temporal dynamics and herbivore stability. It is based on a multi-year dataset from a large forest tree species experiment in southeastern China (BEF-China). Over six years, lepidopteran caterpillars were sampled from a total of 52 experimental plots, covering a tree species gradient from one to 24 tree species. Based on lepidopteran data and repeated estimates of tree growth rate / volume, several indices describing functional diversity / temporal dynamics / stability were determined and related using a series of linear models and path analyses. To better understand underlying patterns, lepidopterans were also separated into food specialists and generalists and analyses were repeated for specialist and generalist subsets. Three hypotheses are tested: (1) temporal dynamics of herbivores are coupled to characteristics and dynamics of tree communities; (2) reduced tree diversity indirectly reduces stability of herbivore communities through reduced stability of tree growth; (3) at the herbivore trophic level, asynchrony drives stability of generalists, while population stability drives stability of specialists. The study finds that the temporal dynamics and stability of herbivores are strongly related to the characteristics and dynamics of tree communities. The relationships between herbivore stability and tree community characteristics and dynamics are mediated by herbivore phylogenetic diversity, asynchrony and population stability, with different relationships between generalist and specialist communities. Higher tree diversity was associated with higher herbivore stability, particularly for specialist herbivores, but this relationship was not mediated by tree growth stability. The authors conclude that lower tree species richness - a common outcome of global change - thus puts herbivore communities at risk.

The study addresses a highly relevant topic and interesting questions based on a robust and interesting dataset. The manuscript is well prepared, demonstrates the relevance of the topic by thoroughly embedding it in the scientific literature, and manages to present a complex topic in a relatively understandable way, making it interesting to the wide readership of Nature Communications. I do have some concerns, mainly related to the design and implementation of the statistical analyses, which I think should be addressed. Some may be based on gaps and ambiguities in the methods sections, which should be revised in some places. Also, although the manuscript is very well written, I would encourage the authors to streamline the discussion, which is currently quite long, e.g. by linking it more closely to the three study hypotheses.

Main concerns:

- The study examines a complex, multi-trophic system based on a set of indices describing the tree community (richness, functional diversity, temporal dynamics/stability) and the herbivore community ("functional" diversity, temporal dynamics/stability). However, it excludes a basic aspect of the herbivore community that I believe is crucial and could explain some of the reported relationships: species richness and abundance of herbivores. Since richness is a fundamental aspect of the study, it should not be reported at only one trophic level. In the presentation of the path analyses, I would suggest including these two parameters at the base of the herbivore community characteristics (adding a third level in Fig. 3). Cf. for example Blüthgen et al. ref. 19 in the present study.
- The separation of generalist and specialist herbivores is a crucial step in this study. Here it is based on the dataset itself, i.e. species (MOTUs) observed to interact with only a few plants are assigned to the specialist taxa (based on a mean pairwise distance approach). I wonder how the overall abundance of a species in the dataset will affect its classification as generalist/specialist. I would expect the 'specialist' category to include not only specialist species, but also rare generalists (which could not be observed on many plants due to small sample sizes). Showing how total abundance and mean pairwise distance are related would be a first step in addressing this potential confounding factor. If the expected relationship is found,

then the results should be reconsidered in this light. For example, I would expect herbivore asynchrony of rare species to be more closely related to herbivore richness stability than that of common species (in the latter case, there is a larger buffer for losing species when variability is higher). In general, an indication of how many individuals are included in each category (generalists, specialists) is missing and should be added. Since the overall dynamics are strongly related to the dynamics of generalists, I assume that generalists form the majority of communities (see also L164ff, L245).

- The dataset is collected within a tree species richness experiment, which includes many plots with only one tree species (24 out of 52). Two of the tree community indices (tree functional diversity, tree asynchrony) are by definition bounded at 0 for these 24 plots, resulting in a set of three highly correlated variables (tree species richness, tree functional diversity, tree asynchrony). To ensure that the observed relationships are not an artefact of these highly constrained variables, I suggest including sensitivity analyses excluding these single-species plots.

- The dataset contains a number of strong outliers (see e.g. Fig. 2). I.e. plots with extremely low specialist herbivore asynchrony (A_N11, A_O27), extremely high specialist herbivore abundance stability (A_T15), extremely high specialist herbivore richness stability (B_V19) and very low tree population stability (B_M29). I suggest including sensitivity analyses excluding these plots.

I have several additional comments that may help to improve the manuscript:

L65f: I would restrict "frequency and severity" to natural disturbances and pest outbreaks. "Frequency of biodiversity loss" is somewhat unclear.

L118: I suggest to quickly introduce the spatial spread of sampling sites here as well. Also, I suggest: "... sampling seasons spread across six years ..."

L132: average population stability of herbivores, right?

L143: I would add the link to Fig. 4

L143ff: Significant positive effects are only found for specialists. The effects for generalists/overall communities are never statistically significant (judged from Fig. 2). So either attribute all effects except for specialist effects as "no effects" or also mention that parts of the relations were negative (although not significant). cf. also L183, where the "no effect" is interpreted as a negative effect.

L149/Figure 4: Could it be an idea to include some more details in Figure 4, i.e. to further distinguish indirect effects through tree FD and tree asynchrony? E.g. by including different fill colours.

L176: This first sentence is a bit unclear. Herbivore diversity was not in the models (only phylogenetic diversity). And why not include the important link to the trees here as well?

L217: I suggest to use a different wording than "dietary niches". Niches are a species trait, which is not affected by the offer of food plants. Maybe change to "food spectrum"?

L225: The "weaker" effect has a larger effect size, though (Fig. 3). It is just the significance which is weaker. I suggest using the term "clearer".

L231-234: I suggest removing this paragraph, as it is merely a repetition of results.

L283ff: It should be made clear that the effect which is hard to explain is the direct effect. As made clear later (L294), the net effect is positive. I think this is crucial here and should be made clear much earlier in this discussion.

L292: I think the argument needs to be explained a bit more, as the mentioned link between abundance and richness is based on absolute numbers, not on stability.

L346: I would mention the 17 sampling events once more here.

L369: Here, the abbreviation MPD is used for a different concept than just a few lines earlier (L362). This confused me a lot, as I was relating to this second definition when trying to understand Fig. 3. The abbreviation MPD should only be used for one concept.

L383: I would add the number of MOTUs that go into each of the categories.

L392: Were these measurements done every year from 2016 to 2021? L408 suggests differently, this should be made clear here. If it is every second year, does it mean that three time points were included? (2016, 2018, 2020)

L401f: Does this mean that individual variation was not included in plots with > 1 species? Unclear formulation.

L417: I think it is important to mention here that Rao's Q is 0 by definition if tree species richness is 1.

L421ff: The temporal level at which these calculations were done is not clear. Was it at the level of sampling events? (n = 17) Or of years? (n = 6) L427 suggests that it is years. If so, it should be made clear from the start that data were pooled per year and plot.

L432: How was species-level stability of trees determined? I could not find this detail. In the data file that was attached for review, I assume the column "pop_tree" to reflect tree population stability. It takes negative values, which I cannot understand based on the formula (3). Please clarify.

L440: Again, please clarify at which temporal resolution this calculation was done.

L442: Why was tree growth asynchrony not determined at individual level, but at species level? From how I understand, this data should be available at individual level. It would have the advantage of decoupling tree species richness 1 from asynchrony 0.

L446: I think it is important to mention here that tree growth asynchrony is 0 by definition if tree species richness is 1.

L458: Why was the interaction site x mean MPD not included?

L463: "... species richness, abundance stability and richness stability ..."

L471: Are the log transformations introduced above also included in the path analyses? If yes, how are they accounted for when aggregating path coefficients (i.e., Fig. 4)?

L489: Can you specify how the non-significant pathways that were excluded were selected?

L706/Figure 2: There are several inconsistencies in this figure. y axes for panels b, d and e, as well as for panels c and f are not the same, although they show the same variables. The y axes labels in panels a and h are missing zero. Also, it is not totally clear whether the linear regression lines are based on the models introduced in L465ff or whether this is a separate set of models (only including one predictor).

L718: The definition of MPD is missing from the legend. Also, what the numbers in the yellow circles are showing is not mentioned (variance explained?).

SI L20 / Fig. S2: is 870 the number of MOTUs that went into the analyses? If yes, why is the number different from the 885 MOTUs introduced at L136?

SI L54 / Table S4: In the overall models, the second predictor variable should be 'herbivore asynchrony', not 'tree asynchrony'. Please check model output tables again, I did not do a systematic check.

Reviewer #2

(Remarks to the Author)

I enjoyed reading the manuscript "Asynchrony and functional diversity couple herbivore and host community dynamics". The authors have tackled a vitally important topic—how community patterns and stability at one trophic level contribute to stability at a higher trophic level—and brought to bear an incredibly impressive dataset from a very large experiment. Moreover, the work is novel, and the study is solidly conducted. I have just three suggestions, and they are focused on improving the framing and analyses. I'd like to thank the authors again for an important contribution to the literature. My suggestions:

First, the study seeks to address the novel and important question of how one trophic level (trees) influence the stability of a higher trophic level (lep herbivores), but much of the text and analyses are focused on how different metrics and aspects of stability relate to each other within just the herbivore community. While it is interesting to explore the relationships between different metrics of herbivore stability at different scales, it felt both a little circular and tangential to the stated goal of examining how tree community dynamics influence herbivore community stability. It felt circular because, unless I'm missing something, it has to be true that the stability of an herbivore community will be determined by the stability and asynchrony of its constituent populations. I suggest either (1) refocusing the manuscript around the effect of the plant community on herbivore stability (and reducing text on the links between different metrics of herbivore stability) or (2) making exploring relationships between different herbivore metrics an explicit secondary goal of the manuscript. The first seems preferable to me, but either could work.

Second, the manuscript is framed around examining the effects of how plant community dynamics influence herbivore community stability, but the authors have relatively weak data on community dynamics. Their data are change in tree biomass over only 6 years. Most of this change seems to be tree growth from saplings to larger trees rather than turnover, which is what most ecologists would consider to be community dynamics. It seems extremely unrealistic to calculate tree population stability or tree asynchrony over just six years. Really what the authors are calculating is variation in tree growth rates, I think, though it's a bit hard to tell from the methods. This could be very relevant for insect herbivores because insect herbivores often perform best when plants grow rapidly, so there is something interesting and important going on here. I just think the authors need to tone down claims of linking tree community dynamics to herbivore community patterns. Instead, the authors measured individual growth patterns and variation in growth patterns. Framing things in terms of community dynamics also obscures the most powerful aspect of the study: the tree species richness treatments. This was a randomized species richness experiment, giving the authors have incredible power to link tree richness to herbivore community patterns and stability. Much of that, however, is lost in the analyses and discussion of tree community dynamics (differences which are downstream from their tree species richness treatments). I suggest focusing the framing and analyses and discussion around the experimental treatment of tree species richness and toning down claims of linking tree community dynamics to herbivore patterns.

Third, the authors use stepwise model selection, but major problems with stepwise model selection have been known for a long time. For example, see Whittingham et al. 2006. Why do we still use stepwise modelling in ecology and behaviour? *Journal of Animal Ecology*. A better approach would be to use AIC to compare a set of candidate models only once. Of course, this may be unwieldy given how many potential models the authors have. This, however, relates to my last two comments. Following my first comment would reduce the number of predictors related to herbivore community metrics. Following my second comment would reduce the number of plant community dynamic predictors. This would simplify the number of potential models down to a more reasonable number of models to compare. Or just fit the full model and focus on the estimated effect sizes.

Finally, the manuscript was well written but a bit hard to read because of the frequent mention of many subtly diversity/stability terms for different scales: abundance stability, richness stability, population stability, and more. Moreover, there are many different facets of stability for any one quantity, and different metrics for each facet. Through the introduction, results, and discussion, it is a little unclear what specifically the authors mean by stability. I suggest simplifying all this, reducing jargon, focusing on the most important aspects of stability for this study, and precisely defining each term early in the manuscript.

Minor comments:

Given the authors' goal of examining stability/variability in trophic interactions and herbivory, two it might be helpful to refer to two recent papers that examine variability in plant-herbivore interactions: Herbivory Variability Network et al. 2023. *Science*.; Wetzel et al. 2023. *Annual Review of Ecology, Evolution, and Systematics*.

The authors use μ / σ as a metric for stability (inverse of CV). This adjusts variability by the mean, but I worry that rare species (with low abundances) will have unrealistically low stability (high variability). This metric will lead to very low values for species that are typically rare because their mean is so low, and it might have nothing to do with stability per se. It might

be helpful to also analyze the data with sigma as the measure of stability to see how that changes results and interpretation. Alternatively, a recent extension of the CV addresses some of these issues and generally makes for a more robust metric: Kvalseth 2016 Journal of Applied Statistics.

I want to end by thanking the authors for their important work. I look forward to seeing it in print!

Reviewer #3

(Remarks to the Author)

Version 1:

Reviewer comments:

Reviewer #1

(Remarks to the Author)

Thank you for this revised version, which I find to have clearly improved and has taken into account all major comments made by the reviewers on the earlier version of this manuscript.

I have few somewhat larger (but not major) points and a set of minor points that could still be addressed.

Larger points:

- Based on comments of both reviewers on the previous version, additional analyses were necessary (herbivore richness, herbivore abundance, version 1 and 2 of stability). Thank you for including these! Clearly, this did not help to reduce the complexity of the manuscript. While I fully support that these additional analyses were made and I think they contribute to the robustness of the results, I would only mention these additional analyses in the main text where really necessary and otherwise relate to them in the methods / supplementary materials. E.g. L184ff, L243ff could be moved to increase clarity.
- It is great that very rare species are excluded for the definition of generalists and specialists, which goes back to a comment I made on the first version. What is however not clear to me now is why all these species were also excluded from the overall analyses. If I see it correctly (L141ff), due to this criterion of 5+ individuals per species, 3844 individuals were excluded from the analyses (more than 20%), which is a lot. Could it be an option to keep these additional individuals in the overall analyses? Or at least lower the threshold of 5? I feel that simply excluding rare species, which apparently make up for a large part of the community, could strongly bias the results.
- I really like the additional information that is now available from Figure 4. One effect that gets really clear now is that higher tree FD is negative for herbivore stability. Although this fact is touched in the discussion (L247ff, L255ff), I think it should be discussed in some more depth. Few more word on the explanation and also on how it relates to other studies would be necessary. And, going in a similar direction, the net effect of tree diversity on herbivore stability is only clearly positive for specialists (Fig. 4). I feel the net negative relation for overall herbivore stability could be acknowledged more. Statements such as L265ff or 273ff do not reflect the results, neither do some conclusions (e.g. L308ff). I think such statements and conclusions should be toned down throughout.

Minor points:

L55: I suggest "... tree functional diversity, tree growth asynchrony, and tree growth population stability."

L84: I suggest "... ecosystems via stability of herbivory."

L88: I suggest: "Herbivore temporal stability is often low in monocultures ...". Generally, sticking to either "stability" or "variability" throughout would increase clarity.

L94: Here it would be good to introduce the concept of "growth variability/stability", which is in other places used to summarise growth asynchrony and growth population stability.

L97: I would replace "species-level" by "population" to keep the wording constant

L119-120: Use "growth variability" or "growth stability" to keep wording constant. Generally, make sure to stick to the same wording throughout, this really helps the readers to follow the text.

L127: herbivore communities (i.e. herbivore phylogenetic diversity) is missing from the sentence.

L128: I feel that Fig. S1 is quite important and really helps to understand the study when being confronted with it for the first time. I would propose to include it as a main figure. Also, in Fig. S1, I think it is a bit unfortunate that the arrow with the label (1) refers to Model 2, the arrow with the label (2) to Model 3, ... It would increase clarity when numbers match.

L149: MPD is mentioned for the first time, the abbreviation should be introduced. To increase clarity, it could help to stick to "herbivore phylogenetic diversity". This term it is used in other places and would also help to reduce confusion between herbivore MPD and host MPD.

L150: What I do not understand about model selection is how you proceeded with the submodels (generalists, specialists). Do these always follow the main model?

L150: When reading the text for the first time, it might be very unclear what herbivore abundance/richness are here all about. As suggested before, I would focus on these alternative models solely or mainly in the methods section.

L156: I would remove “short-term”. Again, this is new terminology that was not used before.

L167: I suggest “... tree functional diversity was negatively related to herbivore asynchrony”, giving the direction of the relation helps.

L241ff: This sentence is somewhat circular with phylogenetic distance being mentioned twice.

L267: H2 explicitly mentions tree growth as a mediator, but it is not touched here.

L276: I suggest “...temporal dynamics of higher trophic levels...”

L283ff: Can you elaborate more on this explanation? I have difficulties to follow, sorry.

L362: The abbreviation was just introduced before, so no need to define it again

L363/387: The exclusion of species with <5 individuals is mentioned twice.

L370f: I would mention at which level weighted MPD were calculated to make clear what this was later used for.

L380, 381, etc: I would clearly write “host MPD” or “herbivore MPD” in all instances throughout the manuscript. This would really help to reduce confusion, although it got already much clearer compared to the first version of the manuscript.

L390: I think here it is missing that this clustering was based on host MPD.

L413ff: To be consistent with equ. 3, I suggest to use TGR_i , $V_{i,t1}$ and $V_{i,t2}$.

L419: Here it is missing what was done with TGR in the end.

L436: Here and in other places. The abbreviation V_1 and V_2 for the model versions are a bit unfortunate, as V_i was also used for the wood volume.

L450: I suggest “... (applying formula 3 or 4).”

L491/500: I would remove abundance and richness here, at is only explained later.

L518/520/523: Add the model terms also for Model 2, Model 3 and Model 4 (equivalent to L514).

Fig. 2: It is still not fully clear to me how these pairwise comparisons that are shown were chosen. Why, for example, it the relation between Herbivore richness stability and Tree species richness missing? Also, how come there is a non-linear relation in Fig. 2d given that these are all linear models (L479ff).

L793: “...(R2) are shown...”

L795: Explanation of the significance level abbreviations missing from the legend.

L805: I guess the effect of herbivore abundance stability on herbivore richness stability was also included? It is missing from the list of variables.

L810: Did the calculations also include non-significant relationships? I think they should. I am just asking because effects via tree population stability are missing from the specialist effect sizes (but they might simply be very small and thus not visible).

L811: Just to be sure, log-transformation was done for both predictor and response variables, right?

Fig. 4: I like the inclusion of the different pathways. What is now somewhat missing is the overall effect, which was represented in the old version. It could still be included. See Figure 3b in Neff et al. (2021) ‘Changes in Plant-Herbivore Network Structure and Robustness along Land-Use Intensity Gradients in Grasslands and Forests’. Science Advances. <https://doi.org/10.1126/sciadv.abf3985> for a possible example.

Reviewer #4

(Remarks to the Author)

Nature communications

Wang et al. 2025—Manuscript # NCOMMS-24-39048A

Overall statement:

This manuscript reports interesting results that connect tree diversity and herbivore community dynamics; it is novel in the intra- and inter-community dynamics across trophic levels. Their results show bottom-up dynamics (loss of tree biodiversity) as important destabilizing factors for specialist herbivores, with implications for global change and biodiversity conservation. The authors make a nice contribution to the long-term and active biodiversity-stability theory debate in ecology. I applaud the substantial effort of the team in this high-investment research (temporally and in caterpillar replication and identification). The paper is quite dense and could use quite a bit of editing to clearly demonstrate how the results contribute to large ecological concepts, rather than get lost in the details of the path analyses; in part, this is due to the authors’ choice to have overall linear models as well as path analysis to understand the patterns (which ends up being difficult to bring together into a cohesive story). A few constructive criticisms:

-Authors should mention (at least in intro) how context-dependency also plays a role in the debate around biodiversity-stability theory

-The results seem to pack too many details in and the main messages get a little lost.

-The authors need to make a better ecological argument in the discussion for how, why, and at what level (population, within vs. between communities, etc.) the asynchrony in tree growth affects the herbivore communities. In general, the argument for why the stability matters (rather than mean differences, etc.) needs to be better justified, for the tree and herbivore metrics.

This will help the paper feel less disconnected and abstract, and more connected to the implications discussed in the introduction and discussion sections.

-The plantation/monoculture areas seem poorly integrated into the intro, results, and discussion. There are a few mentions of the monocultures and it seems like a missed opportunity to connect to human management meaningfully.

-I have a few questions about the “all” category for herbivore types, because on the figures points seem to be a subset (not including the specialists). Please explain/clarify.

-I also have several suggestions for minor changes for improved clarity (see in-line comments below).

In-line comments and section-specific recommendations:

Introduction:

General comments:

Overall, the intro would be strengthened by drawing more directly to diversity-stability theory (and acknowledging the debate/challenges/context dependency within). Because asynchrony is such an important part of the results, it also seems to need more discussion/description in the introduction.

In-line comments

110-111 This section leads the reader to believe they will be comparing among plant growth forms. I would clarify/modify to discuss tree traits more.

134 Final hypothesis could be clarified.

184-196 I recognize this is a response to a reviewer, but it needs to be better connected to the hypotheses or better explained; as is it comes out of the blue and reads exactly like a response to a reviewer, rather than an integrated part of the study.

Results

General comments

I recognize that the authors are trying to strike the balance of enough methods details to give context but leave detailed methods below in the methods section; however, these results are presented relatively confusingly. Please take another readthrough and clarify and simplify. I suggest defining all abbreviations and explaining models more ecologically, rather than statistically (so that the reader does not need to know all the details of the models to understand the important differences). Start out more broad (maybe report overall direct and indirect effects) before getting into the details of each model. I would also suggest connecting the results directly to the hypotheses to better lead the reader through the figures/results. Changes to the topic sentences and/or organization would help.

In-line comments

142 Must use MOTU unabbreviated initially because it is the first mention here.

149 Same comment as above for MPD

Discussion

General comments:

Generally well written and interesting discussion, although I would have liked to see a higher proportion of the overall word count be in this section.

In-line comments

199 If the topic sentence specifies species richness of host trees, it seems slightly strange that the specifics in the next sentences are about functional diversity and tree growth.

204 Yes, this is an important and ecologically intuitive result that should be highlighted

243-248 Same as comment above; needs to be rooted in ecology (or at least directly connected to your hypotheses) to make sense of why you replaced this.

288 Same comment as above, integrate better into the intro to justify

Methods

General comments

The study design (planted saplings across a range of species richnesses) should be mentioned in the intro even though the methods occur afterwards.

In-line comments

325 How old are the saplings at the time of the experiment? If still saplings, I would make sure to say "tree saplings" early on (maybe in the intro justify why they are a particularly good system), in the results, and add at least a sentence in the discussion about how you may expect the results to change as the forest ages.

444 Is it "unrealistically" low stability? Isn't high variability (and low stability) exactly what we would expect from rare species?

507-526 I assume this is a response to a previous reviewer, but to me the four models, rather than structuring a single model based directly on the interactions of interest in the hypotheses, feels disconnected from the main aims of the paper. Maybe having one sentence in the introduction that clarifies that you are specifically interested in the direct, indirect, microclimate, or both (or something that captures all four models) effects would help. This is also not clearly connected to the main results presented in the figures (rather than the supplementals).

Figures

General comments

Figure 1 clearly sets up the "monoculture versus everything else", which isn't the way the analyses were run (more continuous). Consider which is the correct choice for the hypotheses and main messages of the paper.

Figure 2d,e,f. If dark red is "all", why do the lines and data only span the range of the generalist species? Is this just a figure error or does this represent the analyses? Needs a better explanation of "all" either way.

Figure 4 write out functional diversity on the figure legend

Supplemental Figures

General comments

Fig S1. Say which was the final model structure used.

Fig S2. Tree species richness does not appear in this figure, does it? Check to make sure these are the right figures or captions. Also same question as in figure 2: why does "all" not encompass the specialist species?

Fig S3. I would change the title of this to encompass the replacement directly rather than in parentheses.

Fig S4. Same comment as S3

Fig S5 Same comment as Fig 2, and aren't some of these bivariate relationships in Fig 2? (e.g., e?, which also has no label)?

Fig S6 Be clear about how these are different from Fig 3.

Version 2:

Reviewer comments:

Reviewer #1

(Remarks to the Author)

Thank you for this new, revised version. My comments have all been addressed satisfactorily. I have one final remark: In response to a reviewer comment, the conclusions were extended. In my opinion, the conclusions included in lines 353-358 do not reflect the study results, as the destabilization of herbivore communities due to biodiversity loss only holds for specialized herbivores. I suggest to include a little more differentiation here.

Reviewer #4

(Remarks to the Author)

Overall statement:

The authors have sufficiently addressed my previous concerns and the other reviewers'; the manuscript will make a very nice contribution to the field, and I once again commend the authors on their work. Below are a few minor suggestions for small clarifications.

In-line comments:

76 and we lack, rather than but we lack

141-152 May want to consider specifying a, b, c if possible for Figure 1 rather than refer generally to Figure 1 multiple times in these sentences

236 the repercussions of what? Clarify this sentence/few sentences

Asynchrony and functional diversity couple herbivore and host community dynamics

Ming-Qiang Wang, Georg Albert, Carlo L. Seifert, Douglas Chesters, Helge Bruelheide, Jing-Ting Chen, Andréa Davrinche, Sylvia Haider, Shan Li, Yi Li, Goddert von Oheimb, Tobias Proß, Keping Ma, Xiaojuan Liu, Arong Luo, Andreas Schuldt*, Chao-Dong Zhu*

RESPONSE TO REVIEWER 1

Reviewer #1 (Remarks to the Author):

This study investigates the multitrophic relationships linking tree species richness, functional diversity, tree temporal dynamics, herbivore functional diversity, herbivore temporal dynamics and herbivore stability. It is based on a multi-year dataset from a large forest tree species experiment in southeastern China (BEF-China). Over six years, lepidopteran caterpillars were sampled from a total of 52 experimental plots, covering a tree species gradient from one to 24 tree species. Based on lepidopteran data and repeated estimates of tree growth rate / volume, several indices describing functional diversity / temporal dynamics / stability were determined and related using a series of linear models and path analyses. To better understand underlying patterns, lepidopterans were also separated into food specialists and generalists and analyses were repeated for specialist and generalist subsets. Three hypotheses are tested: (1) temporal dynamics of herbivores are coupled to characteristics and dynamics of tree communities; (2) reduced tree diversity indirectly reduces stability of herbivore communities through reduced stability of tree growth; (3) at the herbivore trophic level, asynchrony drives stability of generalists, while population stability drives stability of specialists. The study finds that the temporal dynamics and stability of herbivores are strongly related to the characteristics and dynamics of tree communities. The relationships between herbivore stability and tree community characteristics and dynamics are mediated by herbivore phylogenetic diversity, asynchrony and population stability, with different relationships between generalist and specialist communities. Higher tree diversity was associated with higher herbivore stability, particularly for specialist herbivores, but this relationship was not mediated by tree growth stability. The authors conclude that lower tree species richness - a common outcome of global change - thus puts herbivore communities at risk.

The study addresses a highly relevant topic and interesting questions based on a robust and interesting dataset. The manuscript is well prepared, demonstrates the relevance of the topic by thoroughly embedding it in the scientific literature, and manages to present a complex topic in a relatively understandable way, making it interesting to the wide readership of Nature Communications. I do have some concerns, mainly related to the design and implementation of the statistical analyses, which I think should be addressed. Some may be based on gaps and ambiguities in the methods sections, which should be revised in some places. Also, although the manuscript is very well written, I would encourage the authors to streamline the discussion, which is currently quite long, e.g. by linking it more closely to the three study hypotheses.

Response: We sincerely thank you for your thoughtful comments and constructive feedback on our manuscript. We have carefully addressed all your concerns, particularly regarding the statistical analyses, methodological gaps, and ambiguities. Moreover, we streamlined the discussion to align it more closely with the hypotheses. Please see below for the specific changes according to your comments.

Main concerns:

- The study examines a complex, multi-trophic system based on a set of indices describing the tree community (richness, functional diversity, temporal dynamics/stability) and the herbivore community ("functional" diversity, temporal dynamics/stability). However, it excludes a basic aspect of the herbivore community that I believe is crucial and could explain some of the reported relationships: species richness and abundance of herbivores. Since richness is a fundamental aspect of the study, it should not be reported at only one trophic level. In the presentation of the path analyses, I would suggest including these two parameters at the base of the herbivore community characteristics (adding a third level in Fig. 3). Cf. for example Blüthgen et al. ref. 19 in the present study.

Response: We agree that considering species richness and abundance of herbivores can help to put our results into a broader context. We therefore now consider herbivore abundance and richness in our models to test their potential roles. To avoid overly complex models (as strongly recommended by Reviewer 2 to streamline the models), we decided after careful consideration to replace herbivore MPD with herbivore abundance or herbivore species richness in alternative models, rather than trying to add these predictors on top of the current model structure. Especially since herbivore MPD and herbivore richness were strongly correlated (Pearson's $r = 0.73$, $p < 0.001$) and also abundance was positively associated with herbivore MPD ($r = 0.37$, $p = 0.006$), we considered treating these metrics as largely alternative measures of herbivore diversity to be the most appropriate approach. This is confirmed by the results, which showed that the main pathways connecting host trees and herbivores as well as connections within the herbivore community remained relevant in all three alternative models, although the use of the different measures also revealed interesting new details, (e.g. in contrast to the findings for phylogenetic diversity, direct effects of tree species richness on herbivore abundance and richness remained; Fig. S3, S4; Table S6-11). The corresponding details have been added to the Methods (L503-L505) and Results sections (L184-L188), and the implications of the comparison among the three alternative models are now discussed in the Discussion section (L243-L247).

“To evaluate the role of herbivore abundance and richness in the bottom-up process, we substituted herbivore MPD with herbivore abundance and richness in separate alternative models.”

“Replacing herbivore phylogenetic diversity by herbivore abundance or richness showed overall similar relationships, although direct effects of tree species richness on herbivore abundance and richness remained after accounting for effects of tree functional diversity, tree growth asynchrony and tree population stability (Figs S3, S4; Table S6-11).”

“When herbivore MPD was replaced by herbivore abundance and richness (Fig. S3, S4), the driving role of tree species richness behind these effects became more evident, emphasizing the consequences that biodiversity loss at the host level can have for higher trophic-level diversity and, ultimately, stability.”

- The separation of generalist and specialist herbivores is a crucial step in this study. Here it is based on the dataset itself, i.e. species (MOTUs) observed to interact with only a few plants are assigned to the specialist taxa (based on a mean pairwise distance approach). I wonder how the overall abundance of a species in the dataset will affect its classification as generalist/specialist. I would expect the 'specialist' category to include not only specialist species, but also rare generalists (which could not be observed on many plants due to small sample sizes). Showing how total abundance and mean pairwise distance are related would be a first step in addressing this potential confounding factor. If the expected relationship is found, then the results should be reconsidered in this light. For example, I would expect herbivore asynchrony of rare species to be more closely related to herbivore richness stability than that of common species (in the latter case, there is a larger buffer for losing species when variability is higher). In general, an indication of how many individuals are included in each category (generalists, specialists) is missing and should be added. Since the overall dynamics are strongly related to the dynamics of generalists, I assume that generalists form the majority of communities (see also L164ff, L245).

Response: We agree that rare species can be difficult to evaluate with respect to their host specialization. To avoid potential effects of rare generalists, we therefore now excluded species with fewer than five individuals from our analyses and recalculated all stability and asynchrony metrics based on this more robust data set. Additionally, we examined the correlation between mean abundance and MPD of host specialization for all herbivores ($r = -0.094$) and specialists ($r = -0.160$), and both correlations were weak. In summary, now we are confident that the classification of herbivores into generalists and specialists is not significantly confounded by species abundance, and that effects of rare generalists are not strongly influencing our final results. We have rephased the methods to make readers aware of our approach (L387-L396).

“Before classifying species as either specialists or generalists, we first excluded species with fewer than five individuals. This way, we aimed at eliminating the bias caused by rare species (i.e., that generalists with low abundances are erroneously considered as specialists). Subsequently, we applied hierarchical clustering alongside using two partitioning methods: K-means and partitioning around medoids (PAM)⁵³. This method uses algorithms to group taxa into well-defined 'hard' clusters with minimal prior input, thereby enhancing the objectivity of the classification procedure. Clustering analyses were carried out using two clusters ($k = 2$, i.e., generalists and specialists). The threshold value was then used to separate specialists ($MPD < 0.181$, 103 species) from generalists ($MPD > 0.181$, 140 species).”

- The dataset is collected within a tree species richness experiment, which includes many plots with only one tree species (24 out of 52). Two of the tree community indices (tree functional diversity, tree asynchrony) are by definition bounded at 0 for these 24 plots, resulting in a set of three highly correlated variables (tree species richness, tree functional diversity, tree asynchrony). To ensure that the observed relationships are not an artefact of these highly constrained variables, I suggest including sensitivity analyses excluding these single-species plots.

Response: We have added the suggested sensitivity analysis by running our SEM models after excluding all monocultures. While removing almost half of our data naturally altered the results, the analyses made us confident of our main findings as the main pathways linking host trees and herbivores remained. We discuss the implications in the text (L190-L196):

“The exclusion of monocultures from our analyses expectedly led to weaker effects for some of the observed relationships (e.g. effects of tree species richness on tree growth population stability, relationship between herbivore abundance and richness stabilities; Fig. S8; Table S23-25). However, the main pathways connecting tree species richness with herbivore stability via tree functional diversity, tree growth asynchrony and subsequently herbivore asynchrony and population stability remained strong and significant (Fig. S87).”

and (L287-L289): *“Our observation that effects of tree population stability were reduced even more when removing monocultures from the analyses confirms this assumption.”*

- The dataset contains a number of strong outliers (see e.g. Fig. 2). I.e. plots with extremely low specialist herbivore asynchrony (A_N11, A_O27), extremely high specialist herbivore abundance stability (A_T15), extremely high specialist herbivore richness stability (B_V19) and very low tree population stability (B_M29). I suggest including sensitivity analyses excluding these plots.

Response: Thank you for pointing out these outliers, which helped us to identify potential inconsistencies in our data set. These outliers were apparently caused by the inclusion of extremely rare species. As we now excluded rare species and recalculated all metrics, the data of the above-mentioned plots no longer showed outlier values, obviating the need to conduct additional sensitivity analyses in this case.

I have several additional comments that may help to improve the manuscript:

L65f: I would restrict "frequency and severity" to natural disturbances and pest outbreaks. "Frequency of biodiversity loss" is somewhat unclear.

Response: We deleted "frequency" and kept "biodiversity loss".

L118: I suggest to quickly introduce the spatial spread of sampling sites here as well. Also, I suggest: "... sampling seasons spread across six years ..."

Response: We now added (L123-L125): *“The experiment was conducted at two sites located 4 km apart, comprising randomly distributed study plots with a tree species richness gradient from monoculture to 24 mixtures.”*

L132: average population stability of herbivores, right?

Response: Correct. We now make this clear in the text (L136-L138).

“..., while rarer specialist stability depends more strongly on the average population stability of herbivores.”

L143: I would add the link to Fig. 4

Response: Added as suggested.

L143ff: Significant positive effects are only found for specialists. The effects for generalists/overall communities are never statistically significant (judged from Fig. 2). So either attribute all effects except for specialist effects as "no effects" or also mention that parts of the relations were negative (although not significant). cf. also L183, where the "no effect" is interpreted as a negative effect.

Response: As suggested, we now discuss all effects of tree richness on overall and generalist herbivores as “nonsignificant negative effects”, including the results from L184.

L162-167:

“By contrast, the effects of tree species richness on generalist and, subsequently, also on overall herbivore stability were altogether non-significant, because positive effects via tree growth asynchrony (as well as tree growth population stability) were weaker and counteracted by negative effects via tree functional diversity (Fig. 4) on herbivore population stability (Fig. 2b) and herbivore asynchrony (Fig. 2a).”

and L208-L211:

“...this altogether resulted in less stable abundance dynamics at higher tree species richness via tree functional diversity, even though the direct relationship between tree species richness and herbivore stability was not significant in our linear models.”

L149/Figure 4: Could it be an idea to include some more details in Figure 4, i.e. to further distinguish indirect effects through tree FD and tree asynchrony? E.g. by including different fill colours.

Response: Thank you for this helpful suggestion. We now distinguished the indirect effects through tree FD, tree asynchrony and tree population stability, and updated Figure 4 with different colors.

Fig. 4 Effects of tree species richness on herbivore community stability.

L176: This first sentence is a bit unclear. Herbivore diversity was not in the models (only phylogenetic diversity). And why not include the important link to the trees here as well?

Response: We clarified and rephrased the first sentence of the discussion, focusing more on the important link between herbivore dynamics and their host tree richness (as also suggested by Reviewer 2). It now reads (L199-201):

“Our study unravels how the temporal stability of abundance and species richness of herbivore communities are linked to changes in the species richness of their host tree communities.”

L217: I suggest to use a different wording than "dietary niches". Niches are a species trait, which is not affected by the offer of food plants. Maybe change to "food spectrum"?

Response: Changed accordingly.

L225: The "weaker" effect has a larger effect size, though (Fig. 3). It is just the significance which is weaker. I suggest using the term "clearer".

Response: Thank you for your suggestion. We now changed it accordingly.

L231-234: I suggest removing this paragraph, as it is merely a repetition of results.

Response: We removed the paragraph according to your suggestion, which helped to streamline the discussion.

L283ff: It should be made clear that the effect which is hard to explain is the direct effect. As made clear later (L294), the net effect is positive. I think this is crucial here and should be made clear much earlier in this discussion.

Response: We deleted this part of the discussion as in the updated results, this direct effect was not observed anymore. We took care to highlight the overall net effects clearly in our discussion.

L292: I think the argument needs to be explained a bit more, as the mentioned link between abundance and richness is based on absolute numbers, not on stability.

Response: To streamline the discussion, we now have deleted the argument. Based on the suggestions of Reviewer 2, we focused the discussion more on the interlinkages between host trees and herbivores.

L346: I would mention the 17 sampling events once more here.

Response: Thanks, we mentioned it again as suggested.

L369: Here, the abbreviation MPD is used for a different concept than just a few lines earlier (L362). This confused me a lot, as I was relating to this second definition when trying to understand Fig. 3. The abbreviation MPD should only be used for one concept.

Response: Now we redefined the abbreviation MPD to “the weighted phylogenetic mean pairwise distance” and make it clear in the text what exactly it refers to (caterpillar communities or host tree use).

L383: I would add the number of MOTUs that go into each of the categories.

Response: Now we added the numbers of MOTUs for generalists and specialists (see L142-L143; L394-L396).

“140 were classified as generalist herbivores (accounting for 9063 individuals) and 103 as specialists (with 4943 individuals).”

“The threshold value was then used to separate specialists (MPD < 0.181, 103 species) from generalists (MPD > 0.181, 140 species).”

L392: Were these measurements done every year from 2016 to 2021? L408 suggests differently, this should be made clear here. If it is every second year, does it mean that three time points were included? (2016, 2018, 2020)

Response: Yes, these values were measured every year. The ‘two years’ mentioned in L408 is including previous year and the current year. We now rephased it to make it clear. See L418-L419.

“...where V_{t_1} and V_{t_2} are the tree wood volumes at two consecutive years, t_1 and t_2 , within the sampling period from 2016 to 2021.”

L401f: Does this mean that individual variation was not included in plots with > 1 species? Unclear formulation.

Response: Of course, the individual variation of mixtures was also included. We have rephased the sentence to make this clear (see L413-414).

“...The absolute growth rates (TGR) of individual trees in a plot were calculated as...”

L417: I think it is important to mention here that Rao's Q is 0 by definition if tree species richness is 1.

Response: Now we mention it here to make readers aware of it (see L427-428).

“..., weighted by tree wood volume, and expressed as Rao's Q (note that Rao's Q is 0 in monocultures).”

L421ff: The temporal level at which these calculations were done is not clear. Was it at the level of sampling events? (n = 17) Or of years? (n = 6) L427 suggests that it is years. If so, it should be made clear from the start that data were pooled per year and plot.

Response: Thanks. Now we highlight this at the beginning of the paragraph, clarifying that all data were analyzed per year (i.e. pooling measures within years) (see L431-L433).

“We calculated the temporal stability of herbivores based on the overall abundance and richness of herbivores at plot level and per year (i.e., multiple measures per year were pooled).”

L432: How was species-level stability of trees determined? I could not find this detail. In the data file that was attached for review, I assume the column "pop_tree" to reflect tree population stability. It takes negative values, which I cannot understand based on the formula (3). Please clarify.

Response: Negative values by themselves are not a surprising result given that tree mortality – albeit rarely – can lead to negative growth. However, we reevaluated the way we calculate population stability

following your comment and realized that using negative weights to calculate weighted average population stability is not meaningful and can greatly inflate the effects of negative growth. We therefore switched to using unweighted population stability for both trees and herbivores (see L449-L450), which also reduced the occurrence of negative values to only two plots.

“The plot level-average population stability of tree growth and herbivores were calculated as the sum of all species’ temporal stability indices”

L440: Again, please clarify at which temporal resolution this calculation was done.

Response: We rephrased the sentence to make it clear that all calculations were done on data aggregated per year, as we were interested in annual variability and because tree growth data were measured annually.

L442: Why was tree growth asynchrony not determined at individual level, but at species level? From how I understand, this data should be available at individual level. It would have the advantage of decoupling tree species richness 1 from asynchrony 0.

Response: We really like the idea, but we finally decided against implementing it for a combination of reasons. First, our study builds on previous work that deconstructs community stability in its species asynchrony and population stability components (e.g., Schnabel et al. 2021 Sci Adv <https://www.science.org/doi/full/10.1126/sciadv.abk1643>; Thibaut & Connolly 2013 Ecol Lett 2013 <https://onlinelibrary.wiley.com/doi/full/10.1111/ele.12019>). Moving away from this approach would reduce the comparability of our findings. Second, switching to tree-based analyses might be easily possible for trees, but would not be as straight forward for herbivores, where we would need to focus on herbivore abundances per tree, which can be rather low. This leads to a strong sensitivity to very small changes in abundance. Third, using tree-based analyses would introduce a spatial variability component, which would steer the study in a very different direction and deserves to be investigated properly. We would also lose the diversity aspect of the stability analyses, which was our main focus and seemed to make a lot of sense given that our experiment manipulates diversity specifically. Finally, we knew from our sensitivity analyses that monoculture plots did not drastically alter our findings (L193-L196), hence did not see a strong need to decouple richness 1 from asynchrony 0.

“However, the main pathways connecting tree species richness with herbivore stability via tree functional diversity, tree growth asynchrony and, subsequently, herbivore asynchrony and population stability remained strong and significant (Fig. S8).”

L446: I think it is important to mention here that tree growth asynchrony is 0 by definition if tree species richness is 1.

Response: Now we mention this accordingly to make readers aware (see L461-L463).

“The asynchrony ranges from 0 to 1, representing a gradient from complete synchrony to complete asynchrony (note that asynchrony value is 0 in monocultures).”

L458: Why was the interaction site x mean MPD not included?

Response: We have now included the interaction between site and mean MPD in our linear models. The results indicate that this interaction does not significantly influence the overall outcomes in either model (V_1 or V_2). Specifically, only one model (overall abundance stability, Table S12 for V_1 , Table S19 for V_2) shows a significant effect of tree species richness, but this does not substantially alter the effects of other variables. Although tree richness is retained in the final model with a significant effect, in the other two models (Table S13, S15 for V_1 ; Table S20, S22 for V_2) where the interaction is included, it remains non-significant, and the effects of other variables remain unchanged.

L463: "... species richness, abundance stability and richness stability ..."

Response: Changed accordingly.

L471: Are the log transformations introduced above also included in the path analyses? If yes, how are they accounted for when aggregating path coefficients (i.e., Fig. 4)?

Response: Yes, we also included them in the path analyses. The path coefficients were also based on the models with log-transformed data. Now we highlight this information in the figure captions to make readers aware of this.

L489: Can you specify how the non-significant pathways that were excluded were selected?

Response: Following the suggestions of Reviewer 2, we have changed our modeling procedure and now do not select pathways within individual models, but rather compare multiple alternative model variants in which certain links have been dropped based on pre-defined assumptions (see L495-L506).

“We constructed our initial model according to the current knowledge on mechanism driving biodiversity-stability relationships for plants, insects and ecosystem functions^{11,25,39}. We assumed that tree species richness influences tree functional diversity, tree asynchrony and tree population stability (Fig. S3). Moreover, we assumed that all tree-based predictors can directly affect herbivore metrics, including herbivore MPD/abundance/richness, herbivore asynchrony, herbivore population stability and herbivore stability (Fig. S3). For both trees and herbivores, we hypothesized that functional diversity/MPD can directly affect asynchrony and population stability. To evaluate the role of herbivore abundance and richness in the bottom-up process, we substituted herbivore MPD with herbivore abundance and richness in separate alternative models. We considered covariances between asynchrony and population stability in all models.”

L706/Figure 2: There are several inconsistencies in this figure. y axes for panels b, d and e, as well as for panels c and f are not the same, although they show the same variables. The y axes labels in panels a and h are missing zero. Also, it is not totally clear whether the linear regression lines are based on the models introduced in L465ff or whether this is a separate set of models (only including one predictor).

Response: Thank you for spotting these inconsistencies. We have adapted y axes of all panels of Figure 2 as suggested. For the linear models, we firstly considered multiple predictors. To directly test the relationships between tree species richness and abundance and richness stability of herbivores (separately for all herbivores, generalists and specialists), we then used single linear models, which are only including one predictor (i.e., tree species richness). We have further clarified it in the methods (see L478-L481).

“In addition, we used single linear models to directly test the relationships between tree species richness, abundance and richness stability of herbivores (separately for all herbivores, generalists and specialists).”

L718: The definition of MPD is missing from the legend. Also, what the numbers in the yellow circles are showing is not mentioned (variance explained?).

Response: We now provide the definition of MPD and the meaning of the yellow circles (i.e., explained variance) in the figure legend.

SI L20 / Fig. S2: is 870 the number of MOTUs that went into the analyses? If yes, why is the number different from the 885 MOTUs introduced at L136?

Response: We now optimized the classification of generalists and specialists, so the original figure is not relevant anymore.

SI L54 / Table S4: In the overall models, the second predictor variable should be 'herbivore asynchrony', not 'tree asynchrony'. Please check model output tables again, I did not do a systematic check.

Response: Thank you. We corrected it and checked the full text to avoid the similar errors.

RESPONSE TO REVIEWER 2

Reviewer #2 (Remarks to the Author):

I enjoyed reading the manuscript “Asynchrony and functional diversity couple herbivore and host community dynamics”. The authors have tackled a vitally important topic—how community patterns and stability at one trophic level contribute to stability at a higher trophic level—and brought to bear an

incredibly impressive dataset from a very large experiment. Moreover, the work is novel, and the study is solidly conducted. I have just three suggestions, and they are focused on improving the framing and analyses. I'd like to thank the authors again for an important contribution to the literature. My suggestions:

First, the study seeks to address the novel and important question of how one trophic level (trees) influence the stability of a higher trophic level (lep herbivores), but much of the text and analyses are focused on how different metrics and aspects of stability relate to each other within just the herbivore community. While it is interesting to explore the relationships between different metrics of herbivore stability at different scales, it felt both a little circular and tangential to the stated goal of examining how tree community dynamics influence herbivore community stability. It felt circular because, unless I'm missing something, it has to be true that the stability of an herbivore community will be determined by the stability and asynchrony of its constituent populations. I suggest either (1) refocusing the manuscript around the effect of the plant community on herbivore stability (and reducing text on the links between different metrics of herbivore stability) or (2) making exploring relationships between different herbivore metrics an explicit secondary goal of the manuscript. The first seems preferable to me, but either could work.

Response: Thank you very much for your thoughtful review and constructive feedback. We are pleased that you recognize the contribution of our study to the field. In response to your suggestion, we have revised the manuscript to place greater emphasis on the effects of the host tree communities on herbivore stability, aligning more closely with the primary goal of the study. For instance, we have added and clarified text in sections (e.g., L199-L203) to highlight these relationships. Additionally, we have reduced the focus on the links between different metrics of herbivore stability (e.g. by reconsidering the panels shown in Fig. 2 and reducing the discussing on the topic in L279-L299), but did not completely remove the exploration of within-herbivore community metrics, as we consider this part an important contribution to shedding light on the “black box” of trophic interactions. Specifically, it allows us to show the pathways via which host tree effects ultimately affect herbivore stability. We believe the changes make the manuscript more cohesive and aligned with the stated objectives.

Second, the manuscript is framed around examining the effects of how plant community dynamics influence herbivore community stability, but the authors have relatively weak data on community dynamics. Their data are change in tree biomass over only 6 years. Most of this change seems to be tree growth from saplings to larger trees rather than turnover, which is what most ecologists would consider to be community dynamics. It seems extremely unrealistic to calculate tree population stability or tree asynchrony over just six years. Really what the authors are calculating is variation in tree growth rates, I think, though it's a bit hard to tell from the methods. This could be very relevant for insect herbivores because insect herbivores often perform best when plants grow rapidly, so there is something interesting and important going on here. I just think the authors need to tone down claims of linking tree community dynamics to herbivore community patterns. Instead, the authors measured individual growth patterns and variation in growth patterns. Framing things in terms of community dynamics also obscures the most powerful aspect of the study: the tree species richness treatments. This was a randomized species richness experiment, giving the authors have incredible power to link tree richness to herbivore community patterns and stability. Much of that, however, is lost in the analyses and discussion of tree community dynamics (differences which are downstream from their tree species richness treatments). I suggest focusing the framing and analyses and

discussion around the experimental treatment of tree species richness and toning down claims of linking tree community dynamics to herbivore patterns.

Response: Thank you for your insightful suggestions. We have addressed your concerns, toning down the claims regarding community dynamics of host trees, and instead focusing more on the (indirect) linkages between tree species richness and herbivore stability. Specifically, 1) We revised the manuscript title to emphasize the importance of tree species richness as the central focus of the study. 2) We rewrote the discussion of tree community metrics and their influence on herbivore patterns, instead highlighting tree species richness as the primary driver and taking care throughout the manuscript that our results are not mistaken for long-term, decadal plant-herbivore community dynamics (e.g. see L156, L229-L261). 3) We clarified throughout the manuscript that our data primarily reflect variation in tree growth rates rather than longer-term community dynamics (e.g. changed tree dynamics to tree growth variability) in the traditional sense, ensuring that our framing aligns with the scope and nature of the data. These revisions help to better align the manuscript with its strongest aspects, particularly the experimental design and the role of tree species richness in driving herbivore community patterns and stability. We believe these changes enhance the clarity and focus of the study. We kept the terminology of asynchrony and population stability for trees, however, because these are standard terms easily understandable to readers familiar with temporal community analyses and because these terms have also been used in similar temporal context in other studies (e.g. Schnabel et al. 2021 Sci Adv <https://www.science.org/doi/full/10.1126/sciadv.abk1643>), ensuring that our results can be understood in the context of related studies.

Third, the authors use stepwise model selection, but major problems with stepwise model selection have been known for a long time. For example, see Whittingham et al. 2006. Why do we still use stepwise modelling in ecology and behaviour? Journal of Animal Ecology. A better approach would be to use AIC to compare a set of candidate models only once. Of course, this may be unwieldy given how many potential models the authors have. This, however, relates to my last two comments. Following my first comment would reduce the number of predictors related to herbivore community metrics. Following my second comment would reduce the number of plant community dynamic predictors. This would simplify the number of potential models down to a more reasonable number of models to compare. Or just fit the full model and focus on the estimated effect sizes.

Response: Thank you for highlighting the issues with stepwise model selection and providing a constructive alternative. We have revised our model selection procedure in line with your suggestions. Specifically, we proposed four candidate model structures based on our hypotheses and prior knowledge. We then selected the best model structure by comparing model fit statistics (e.g., AIC, see Table S1, S2). L509-L525:

“Moreover, to assess the level of support for different ways in which host tree-based metrics influence herbivore community dynamics, we compared four alternative model structures that successively removed interactions between different levels of host tree and herbivore community data (Fig. S1): First, we considered the initial model with all potential links included (Model 1). Second, we assumed that effects are entirely mediated by stability components, except for effects such as microclimate buffering that may remain in tree species richness. Therefore, we removed the direct connections between trees and herbivore

abundance and richness stability, except for effects of tree species richness. Third, we assumed that there are no such effects of tree species richness, so we additionally excluded the direct pathways between tree species richness and herbivore stability metrics. Fourth, we assumed that tree species richness only has indirect effects via tree stability components and functional composition. We hence removed the links between tree species richness and herbivore MPD/abundance/richness and population stability. Model fit statistics were summarized (Table S1, S2). The best-fitting model was selected based on the lowest AIC value⁶². We used bootstrapped P values based on 1000 bootstrap draws to ensure robust results⁶³.

Finally, the manuscript was well written but a bit hard to read because of the frequent mention of many subtly diversity/stability terms for different scales: abundance stability, richness stability, population stability, and more. Moreover, there are many different facets of stability for any one quantity, and different metrics for each facet. Through the introduction, results, and discussion, it is a little unclear what specifically the authors mean by stability. I suggest simplifying all this, reducing jargon, focusing on the most important aspects of stability for this study, and precisely defining each term early in the manuscript.

Response: Thank you for this valuable suggestion. We agree that reducing jargon will make it easier for readers to understand the core message of our study. In response, we have streamlined the terminology throughout the manuscript, focusing on the most important aspects of stability relevant to this study and reducing the use of overly technical terms. For instance, temporal dynamics, among other adjustments. Additionally, we have provided precise definitions of key terms, such as abundance stability, richness stability, and population stability, early in the manuscript to ensure clarity and consistency, so that readers can better follow the necessary distinctions we make among different components of the host tree and herbivore community data (e.g. L82-L83, L95-L96). We hope that the revised text strikes a balance between avoiding overly complex jargon and nevertheless providing well-defined definitions of necessary key terms. We believe these changes will significantly improve the readability and accessibility of our work.

Minor comments:

Given the authors' goal of examining stability/variability in trophic interactions and herbivory, two it might be helpful to refer to two recent papers that examine variability in plant-herbivore interactions: Herbivory Variability Network et al. 2023. Science.; Wetzal et al. 2023. Annual Review of Ecology, Evolution, and Systematics.

Response: Thanks for your suggestion. We now included the suggested papers in our manuscript (see L82-84).

“The temporal stability (i.e. invariability in time at community and population level) of herbivore abundance can substantially alter ecosystems via herbivory^{17,18}.”

The authors use μ / σ as a metric for stability (inverse of CV). This adjusts variability by the mean, but I

worry that rare species (with low abundances) will have unrealistically low stability (high variability). This metric will lead to very low values for species that are typically rare because their mean is so low, and it might have nothing to do with stability per se. It might be helpful to also analyze the data with sigma as the measure of stability to see how that changes results and interpretation. Alternatively, a recent extension of the CV addresses some of these issues and generally makes for a more robust metric: Kvalseth 2016 Journal of Applied Statistics.

Response: We have now additionally checked the results based on the metric proposed by Kvalseth (2016) in our analyses and found that the results are largely consistent with those based on the inverse of CV. This is probably due to the fact that we now excluded rare species (n < 5) from our dataset, thereby minimizing the potential influence of low-abundance species on stability metrics. This approach strengthens the robustness of our conclusions by ensuring that our results are not disproportionately driven by species with inherently low stability due to low abundance. Thank you for suggesting this improvement, which has enhanced the reliability of our findings. See L443-L447 in methods, L186-L188 in results, and Fig. S5-7, Table S16-22.

“Moreover, to further eliminate the potential bias caused by rare species (e.g., unrealistically low stability), we also utilized an alternative stability metric from Kvålseth et al.⁵⁸

$$\text{stability } (V_2) = \frac{\sqrt{\sigma^2 + \mu^2}}{\sigma^2},$$

“In addition, our findings remained highly consistent when an alternative stability metric (V₂) was used in our models (Fig. S2d-f, S5-7; Table S16-22).

I want to end by thanking the authors for their important work. I look forward to seeing it in print!

Response: Thanks again for your constructive comments to help us to further improve the manuscript!

RESPONSE TO REVIEWER 3

Reviewer #3 (Remarks to the Author):

Response: Thank you for supporting the review process. We have thoroughly considered all comments, which greatly enhanced the clarity and robustness of our manuscript.

Asynchrony and functional diversity couple herbivore community dynamics to host plant diversity

Ming-Qiang Wang, Georg Albert, Carlo L. Seifert, Douglas Chesters, Helge Bruelheide, Jing-Ting Chen, Andréa Davrinche, Sylvia Haider, Shan Li, Yi Li, Goddert von Oheimb, Tobias Proß, Keping Ma, Xiaojuan Liu, Arong Luo, Andreas Schuldt*, Chao-Dong Zhu*

RESPONSE TO REVIEWER 1

Reviewer #1 (Remarks to the Author):

Thank you for this revised version, which I find to have clearly improved and has taken into account all major comments made by the reviewers on the earlier version of this manuscript.

I have few somewhat larger (but not major) points and a set of minor points that could still be addressed.

Response: We sincerely thank the reviewer for the additional suggestions, which helped us to further improve the clarity and quality of our work. Below, we address each of the points in detail.

Larger points:

- Based on comments of both reviewers on the previous version, additional analyses were necessary (herbivore richness, herbivore abundance, version 1 and 2 of stability). Thank you for including these! Clearly, this did not help to reduce the complexity of the manuscript. While I fully support that these additional analyses were made and I think they contribute to the robustness of the results, I would only mention these additional analyses in the main text where really necessary and otherwise relate to them in the methods / supplementary materials. E.g. L184ff, L243ff could be moved to increase clarity.

Response: We agree that streamlining the presentation of the additional analyses improves clarity. Following your suggestion, we have now kept references to these analyses in the main text only where they are essential for interpreting the results (see L155-L200). Especially, in all other cases, we moved the methods and results of the sensitivity analyses, as well as the corresponding figures and tables, into the supplementary information to keep the clarity of the main text.

- It is great that very rare species are excluded for the definition of generalists and specialists, which goes back to a comment I made on the first version. What is however not clear to me now is why all these species were also excluded from the overall analyses. If I see it correctly (L141ff), due to this criterion of 5+ individuals per species, 3844 individuals were excluded from the analyses (more than 20%), which is a lot. Could it be an option to keep these additional individuals in the overall analyses? Or at least lower the threshold of 5? I feel that simply excluding rare species, which apparently make up for a large part of the community, could strongly bias the results.

Response: Thank you for raising this important point and for revisiting your earlier comment. Following the previous reviewer comments, we applied the ≥ 5 individuals threshold across all analyses to ensure comparability among generalist, specialist, and overall herbivore datasets, and to reduce the influence of potentially vagrant individuals. However, we agree that it is important to show whether and how the

exclusion of rare species from the overall analyses may influence the results. Therefore, we have now re-run the overall analyses including all individuals (i.e., without the ≥ 5 threshold) as an additional sensitivity analysis and present the results in the Supplementary Materials (new Fig. S7 and Tables S23-24). The main conclusions remain qualitatively unchanged (e.g. negative effects of tree FD on herbivore stability via herbivore asynchrony and positive effects of tree species richness on herbivore stability through tree asynchrony and herbivore MPD), suggesting that the overall patterns are robust to the exclusion of extremely rare species (see L187-L191).

Fig. S7 Sensitivity analysis for effects of tree diversity on herbivore community stability via bottom-up regulation including all herbivore individuals. Potential effects of tree species richness, tree functional diversity (FD), species asynchrony and population stability on community stability of herbivore abundance and richness through mean herbivore abundance weighted phylogenetic diversity (MPD), species asynchrony and population stability for **a** overall herbivores using stability based on the inverse of the coefficient of variation ($\chi^2 = 5.65$, $DF = 11$, $P = 0.890$), **b** overall herbivores using stability following Kvålseth et al. ($\chi^2 = 11.19$, $DF = 11$, $P = 0.468$) based on path model results (see Tables S23-34 for full results). Blue arrows indicate positive effects, red arrows show negative effects ($p \leq 0.1$), grey arrows show non-significant pathways ($p > 0.1$). Arrow width was scaled by the standardized path coefficients. The proportion of variance (R^2) are shown in yellow circles. Note that tree species richness, population stability, abundance stability and richness stability of herbivores were log-transformed. Significance levels: $p < 0.1$ (·), $p < 0.05$ (*), $p < 0.01$ (**), $p < 0.001$ (***)

- I really like the additional information that is now available from Figure 4. One effect that gets really clear now is that higher tree FD is negative for herbivore stability. Although this fact is touched in the discussion (L247ff, L255ff), I think it should be discussed in some more depth. Few more word on the explanation and also on how it relates to other studies would be necessary. And, going in a similar direction, the net effect of tree diversity on herbivore stability is only clearly positive for specialists (Fig. 4). I feel the net negative relation for overall herbivore stability could be acknowledged more. Statements such as L265ff or 273ff do not reflect the results, neither do some conclusions (e.g. L308ff). I think such statements and conclusions should be toned down throughout.

Response: Thank you for highlighting these important points. We agree that the negative relationship

between tree functional diversity and herbivore stability, as well as the net negative relation for overall herbivore stability, deserves more explicit discussion. We have now expanded the relevant discussion sections (see L287-L297) to provide a more in-depth interpretation of the negative tree FD effect, including potential ecological mechanisms and how our findings compare with previous studies:

“...Therefore, the consistent negative effects of tree functional diversity on both overall and generalist herbivore stability suggest that increased functional differentiation among trees creates greater heterogeneity in resource traits, which in turn destabilizes herbivore populations when they depend on specific host functional types. Similar findings have been reported in other multitrophic studies, where high functional diversity of basal resources increased variability in consumer communities^{13,15}. Importantly, while tree species richness had a stabilizing effect on specialist herbivores, the net effect of tree diversity on overall herbivore stability was weak or even negative, highlighting that positive biodiversity–stability relationships may not uniformly extend across all consumer guilds...”

We also explicitly acknowledge in the revised text that the net effect of tree diversity on herbivore stability is only clearly positive for specialists, while it is neutral to negative for overall herbivores. Correspondingly, we have toned down and reworded statements in the Results, Discussion, and Conclusions (see L293-L297) to more accurately reflect the patterns shown in Fig. 4.

Minor points:

L55: I suggest “... tree functional diversity, tree growth asynchrony, and tree growth population stability.”

Response: Changed accordingly.

L84: I suggest “... ecosystems via stability of herbivory.”

Response: Changed as suggested.

L88: I suggest: “Herbivore temporal stability is often low in monocultures ...”. Generally, sticking to either “stability” or “variability” throughout would increase clarity.

Response: Changed as suggested, we now used “stability” throughout the full text.

L94: Here it would be good to introduce the concept of “growth variability/stability”, which is in other places used to summarise growth asynchrony and growth population stability.

Response: Thank you for the helpful suggestion. We have now clarified the concept of “growth variability/stability” in the revised sentence, highlighting its role in summarising both growth asynchrony and population-level growth stability (see L99-L102):

“For tree communities, the temporal stability in growth and productivity has recently received increasing attention^{14,24}. In general, community stability can be partitioned into species asynchrony, i.e. a temporal misalignment of species dynamics, and the stability of species’ populations themselves¹⁴”

L97: I would replace “species-level” by “population” to keep the wording constant

Response: Changed as suggested.

L119-120: Use “growth variability” or “growth stability” to keep wording constant. Generally, make sure to stick to the same wording throughout, this really helps the readers to follow the text.

Response: We now used “stability” in the full text for clarity.

L127: herbivore communities (i.e. herbivore phylogenetic diversity) is missing from the sentence.

Response: We now provided relevant information to make reader to follow the text (see L136-L140):

“Specifically, we used linear models and path analysis to test the pathways that directly and indirectly (via tree and herbivore community asynchrony and population stability, and via herbivore phylogenetic diversity) connect tree species richness and its effects on tree growth to the temporal stability of herbivore abundance and species richness”

L128: I feel that Fig. S1 is quite important and really helps to understand the study when being confronted with it for the first time. I would propose to include it as a main figure. Also, in Fig. S1, I think it is a bit unfortunate that the arrow with the label (1) refers to Model 2, the arrow with the label (2) to Model 3, ... It would increase clarity when numbers match.

Response: We appreciate this suggestion. We have now integrated the information from former Fig. S1 into Fig. 1 of the main text. Moreover, we also reduced the numbering of arrows to improve clarity.

Fig. 1 Conceptual figure and initial path model structure illustrating the determining mechanisms of herbivore community stability. Herbivore abundance asynchrony among species and community stability are coupled with the diversity and temporal growth rate stability of their host tree communities (Hypothesis 1), with herbivore community stability ultimately **a** destabilized by low tree species richness and **b** stabilized by high tree species richness (Hypothesis 2). Specifically, in a monocultures and species-poor mixtures, the pronounced fluctuations in tree growth may lead to reduced asynchrony among herbivore species, thereby destabilizing the herbivore community. Conversely, in **b** more diverse mixtures, complementary dynamics resulting from the asynchrony among tree species could contribute to the stabilization of plant communities. This stabilization effect extends to herbivore communities, promoting greater asynchrony among herbivore species and ultimately enhancing the stability of the herbivore community (Hypothesis 3). These relationships were investigated with **c** path models. Structure based on theoretical expectations and correlations among herbivore- and tree-based variables: mean phylogenetic distance (MPD), species asynchrony, population stability of herbivores; tree functional diversity (FD), species asynchrony, population stability of trees. Arrows indicate expected causal relationships. Blue lines are covariances retained in the path models. We assessed with four alternative models how direct vs. indirect effects of host tree-based metrics influence herbivore community dynamics. Model 1: both direct and indirect pathways from trees to herbivore stability; Model 2: restricting tree growth effects to indirect pathways via herbivore population stability and asynchrony, except for tree species richness; Model 3: assuming even species richness acts only indirectly; Model 4: assuming that tree diversity influences herbivores solely through effects on tree functional diversity, asynchrony, and population stability. Model 4 was selected for our final analyses.

L149: MPD is mentioned for the first time, the abbreviation should be introduced. To increase clarity, it could help to stick to “herbivore phylogenetic diversity”. This term it is used in other places and would also help to reduce confusion between herbivore MPD and host MPD.

Response: Thank you for pointing this out. We have revised the sentence to introduce the abbreviation “MPD” upon its first mention. Moreover, we now used “herbivore phylogenetic diversity” to reduce confusion between herbivore MPD and host MPD.

L150: What I do not understand about model selection is how you proceeded with the submodels (generalists, specialists). Do these always follow the main model?

Response: Yes. We have clarified that the model structure selected for overall herbivores was also applied to generalists and specialists to ensure comparability and interpretability across groups (see L562-564): *“The same model structure selected for overall herbivores was applied to generalists and specialists to ensure comparability across groups.”*

L150: When reading the text for the first time, it might be very unclear what herbivore abundance/richness are here all about. As suggested before, I would focus on these alternative models solely or mainly in the methods section.

Response: Thanks for the suggestion. We have streamlined the content in the main text and refer readers to the Methods section for details on our alternative models that we used as sensitivity analyses (See Supplementary L26-L33):

“To evaluate the role of herbivore abundance and richness in the bottom-up process, we substituted herbivore MPD with herbivore abundance or richness in separate alternative models. While we assumed that

herbivore MPD best captures diversity effects because it combines abundance-weighted differences in not only species richness, but phylogenetic (and thereby functional) differentiation among species², models using herbivore abundance or richness may be helpful in evaluating the contribution of overall herbivore density and the mere number of species, respectively.”

L156: I would remove “short-term”. Again, this is new terminology that was not used before.

Response: Now it was removed as suggested.

L167: I suggest “... tree functional diversity was negatively related to herbivore asynchrony”, giving the direction of the relation helps.

Response: Changed as suggested.

L241ff: This sentence is somewhat circular with phylogenetic distance being mentioned twice.

Response: Thank you for pointing this out. We have revised the sentence to remove the circular phrasing and to clarify the proposed mechanism linking temporal resource variability to herbivore phylogenetic diversity (see L259-L263):

“Tree growth asynchrony promoted herbivore phylogenetic diversity of both generalists and specialists, suggesting that temporal stability in the availability of food resources might provide more niche opportunities that favor the coexistence of more distantly related herbivore species and, in consequence, functionally more diversified consumer communities with more distant phylogenetic relationships³⁷”

L267: H2 explicitly mentions tree growth as a mediator, but it is not touched here.

Response: We now have adapted H2 and discussion to make this clearer (see L148-L149; L301-L304):

“... the loss of tree diversity indirectly (via tree growth) destabilizes herbivore communities (Fig. 1).”

“The resulting overall direct and indirect effects via tree growth of tree species richness on specialist herbivore community stability were clearly positive, partly supporting our expectation (H2) and underlining the crucial role that tree species richness may play in preventing outbreaks of often specialized insect pests⁴⁴”

L276: I suggest “...temporal dynamics of higher trophic levels...”

Response: Changed as suggested.

L283ff: Can you elaborate more on this explanation? I have difficulties to follow, sorry.

Response: Thank you for your comment. We have now clarified this explanation to better communicate the underlying mechanism. Specifically, we explain that in mixed-species plots, community-level abundance stability is primarily driven by asynchronous fluctuations among host-specific herbivores, rather than by the stability of individual populations. This interpretation is consistent with our results, where asynchrony was the main driver of stability for both generalists and specialists, despite our expectation (H₃) that population stability would play a stronger role in the latter (see L319-327):

“This deviation may be explained by the fact that many of our study plots were mixtures of different tree species, which hosted distinct herbivore assemblages. In such mixed-species plots, mean plot-level herbivore abundance (calculated across all herbivore species) does not necessarily rely on stable populations of each species. Instead, it can be maintained when fluctuations in abundance are asynchronous among species—even among specialists—because multiple host tree species support different herbivore populations. These compensatory dynamics buffer total herbivore abundance over time, reducing the apparent influence of population stability on plot-level stability²⁵”

L362: The abbreviation was just introduced before, so no need to define it again

Response: Done.

L363/387: The exclusion of species with <5 individuals is mentioned twice.

Response: Thank you for pointing this out. We have removed the redundant sentence and now mention the exclusion of species with <5 individuals only once (see L432-433):

“Before classifying species as either specialists or generalists, we first excluded species with fewer than five individuals.”

L370f: I would mention at which level weighted MPD were calculated to make clear what this was later used for.

Response: We now specified the level at which we calculated weighted MPD to promote clarity (see L414-L416):

“We then calculated abundance weighted mean pairwise phylogenetic distance (MPD) for sets of the phylogenetically-placed herbivore MOTUs at community level.”

L380, 381, etc: I would clearly write “host MPD” or “herbivore MPD” in all instances throughout the manuscript. This would really help to reduce confusion, although it got already much clearer compared to the first version of the manuscript.

Response: Thank you for highlighting this important point. To improve clarity and consistency, we have revised the manuscript to explicitly distinguish between “host MPD” and “herbivore MPD” in all relevant instances.

L390: I think here it is missing that this clustering was based on host MPD.

Response: Thanks. We have clarified this clustering was based on host MPD.

L413ff: To be consistent with equ. 3, I suggest to use TGR_i , $V_{i,t1}$ and $V_{i,t2}$.

Response: Changed for consistence.

L419: Here it is missing what was done with TGR in the end.

Response: We provided an additional sentence to explain the further calculations based on TGR (see L464-465):

“ TGR_i was used for the calculation of tree asynchrony and population stability.”

L436: Here and in other places. The abbreviation V_1 and V_2 for the model versions are a bit unfortunate, as V_i was also used for the wood volume.

Response: Thank you for this helpful comment. We agree that the overlapping abbreviations could cause confusion. We now call it stability (the inverse of the coefficient of variation) in the main text, and refer to stability following Kvålseth et al. for the alternative measure in supplementary materials.

L450: I suggest "... (applying formula 3 or 4)."

Response: Changed as suggested.

L491/500: I would remove abundance and richness here, as it is only explained later.

Response: We have removed abundance and richness here for clarity.

L518/520/523: Add the model terms also for Model 2, Model 3 and Model 4 (equivalent to L514).

Response: Added as suggested.

Fig. 2: It is still not fully clear to me how these pairwise comparisons that are shown were chosen. Why, for example, is the relation between Herbivore richness stability and Tree species richness missing? Also, how come there is a non-linear relation in Fig. 2d given that these are all linear models (L479ff).

Response: For the selection of pairwise comparisons, we aimed to highlight those relationships that also emerged as important pathways in the SEM and/or were particularly important for the discussion of the results. We did not add the relationship between tree species richness and herbivore richness stability since the latter was strongly related to herbivore abundance stability and the relationships therefore similar to the ones in panel c. We have now corrected Fig. 2d to display the appropriate linear relationship consistent with our linear models (and have also updated the corresponding supplementary figure).

L793: "...(R2) are shown..."

Response: changed accordingly.

L795: Explanation of the significance level abbreviations missing from the legend.

Response: Thank you for the comment. We have now added an explanation of the significance level abbreviations (see L846-L847):

"Significance levels: $p < 0.1$ (.), $p < 0.05$ (), $p < 0.01$ (**), $p < 0.001$ (**)."*

L805: I guess the effect of herbivore abundance stability on herbivore richness stability was also included? It is missing from the list of variables.

Response: Yes. We now listed herbivore abundance stability as variables for herbivore richness stability.

L810: Did the calculations also include non-significant relationships? I think they should. I am just asking because effects via tree population stability are missing from the specialist effect sizes (but they might simply be very small and thus not visible).

Response: We did not include non-significant pathways in the original calculation, as most of these pathways showed very high uncertainty (large standard errors). Displaying them in the main figure alone could therefore have been misleading for the interpretation of the overall effects. To ensure transparency, we now provide summaries of both approaches: (i) effect sizes based only on significant pathways (main text), and (ii) effect sizes including non-significant pathways (Supplementary Fig. S9). To better capture the

uncertainty of non-significant path ways, we decided to bootstrap these effects. The corresponding details have been added to the Methods (see L566-569) and Supplementary Results (see L60-63):

In Methods:

“The effect sizes of tree species richness on herbivore richness stability were then summarized. We additionally calculated effect sizes including non-significant pathways and assessed their variability using 1,000 bootstrap draws.”

In Supplementary Results:

“When non-significant pathways were included, effects of tree asynchrony on all herbivores and of tree population stability on generalists were more prominent, but characterized by high uncertainty (Fig. S9X).”

Fig. S9 Bootstrapped effects of tree species richness on herbivore richness stability including all pathways (significant and non-significant). Bars show summed effects of tree species richness on the abundance and richness stability of all, generalist and specialist herbivores, respectively. Error bars indicate 95% bootstrap confidence intervals. Effect sizes were calculated by summing indirect effects of tree species richness via tree functional diversity, tree asynchrony, tree population stability, herbivore phylogenetic diversity, herbivore asynchrony, herbivore population stability, and herbivore abundance stability. The different colors show effects of tree species on herbivore stability via tree functional diversity, tree asynchrony, and tree population stability, respectively. Effect sizes were calculated as the product of standardized path coefficients connecting each predictor with herbivore components, summed over the individual predictors of each component for positive and negative effects on herbivore stability metrics, respectively. Note that tree species richness, population stability, abundance stability and richness stability of herbivores were log-transformed. Stability measures are based on the inverse of the coefficient of variation (eqn. 3).

L811: Just to be sure, log-transformation was done for both predictor and response variables, right?

Response: Yes, we log-transformed for both predictor and response variables. We have specified this in the Methods (see L524-L525).

Fig. 4: I like the inclusion of the different pathways. What is now somewhat missing is the overall effect, which was represented in the old version. It could still be included. See Figure 3b in Neff et al. (2021) ‘Changes in Plant-Herbivore Network Structure and Robustness along Land-Use Intensity Gradients in Grasslands and Forests’. Science Advances. <https://doi.org/10.1126/sciadv.abf3985> for a possible example.

Response: We now have included again the different pathways and overall effect in the figure.

Fig. 4 Effects of tree species richness on herbivore richness stability. Bars show summed effects of tree species richness on the abundance and richness stability of all, generalist and specialist herbivores, respectively. Effect sizes were calculated by summing indirect effects of tree species richness via tree functional diversity, tree asynchrony, tree population stability, herbivore phylogenetic diversity, herbivore asynchrony, herbivore population stability, and herbivore abundance stability. The different colors show effects of tree species on herbivore stability via tree functional diversity, tree asynchrony, and tree population stability, respectively. Effect sizes were calculated as the product of standardized path coefficients connecting each predictor with herbivore components, summed over the individual predictors of each component for positive and negative effects on herbivore stability metrics, respectively. Black T-shaped lines indicate the total effects of tree species richness on herbivore stability metrics. Note that tree species richness, population stability, abundance stability and richness stability of herbivores were log-transformed. Stability measures are based on the inverse of the coefficient of variation (eqn. 3). See Supplementary Results and Fig. S9 for effects including non-significant pathways.

RESPONSE TO REVIEWER 4

Reviewer #4 (Remarks to the Author):

Overall statement:

This manuscript reports interesting results that connect tree diversity and herbivore community dynamics; it is novel in the intra- and inter-community dynamics across trophic levels. Their results show bottom-up dynamics (loss of tree biodiversity) as important destabilizing factors for specialist herbivores, with implications for global change and biodiversity conservation. The authors make a nice contribution to the long-term and active biodiversity-stability theory debate in ecology. I applaud the substantial effort of the team in this high-investment research (temporally and in caterpillar replication and identification). The paper is quite dense and could use quite a bit of editing to clearly demonstrate how the results contribute to large ecological concepts, rather than get lost in the details of the path analyses; in part, this is due to the authors' choice to have overall linear models as well as path analysis to understand the patterns (which ends up being difficult to bring together into a cohesive story). A few constructive criticisms:

Response: Thank you for the constructive suggestions regarding the clarity and cohesion of our narrative. In the revised manuscript, we have streamlined the presentation of the results by reducing unnecessary detail from the main text, focusing more explicitly on how our findings contribute to broader ecological concepts.

-Authors should mention (at least in intro) how context-dependency also plays a role in the debate around biodiversity-stability theory

Response: We agree with the reviewer that context-dependency is a key factor contributing to the complexity and ongoing debate surrounding biodiversity–stability relationships. We have now addressed this in the Introduction by highlighting how environmental conditions such as climate can modulate the strength and direction of biodiversity–stability effects (see L74-L77):

“... This is because, for one, biodiversity–stability relationships may vary with environmental (e.g. climate) conditions and across spatial and temporal scales^{7,12,13}, but we currently lack insights from many ecosystems.”

-The results seem to pack too many details in and the main messages get a little lost.

Response: we have streamlined the presentation of the results to strengthen the core message. In particular, we have moved all sensitivity analysis results to the Supplementary Materials, which helps to improve clarity and keep the focus on the main findings.

-The authors need to make a better ecological argument in the discussion for how, why, and at what level (population, within vs. between communities, etc.) the asynchrony in tree growth affects the herbivore communities. In general, the argument for why the stability matters (rather than mean differences, etc.) needs to be better justified, for the tree and herbivore metrics. This will help the paper feel less disconnected and abstract, and more connected to the implications discussed in the introduction and discussion sections.

Response: We thank the reviewer for pointing this out. We have now expanded our main text of how and why tree growth asynchrony affects herbivore communities, and at what level these effects are likely to

operate. Moreover, we have provided a clearer ecological argument for why stability, rather than mean differences alone, is critical to consider by incorporating a more mechanistic explanation (see L230-L242):

“Tree growth asynchrony could stabilize herbivore communities by buffering resource availability at the population level, reducing competition within communities³². Moreover, our study highlights the added value of expanding biodiversity and ecosystem functioning relationships by explicitly considering stability relationships. While high average biomass production may reflect favorable conditions in some years, it may mask substantial interannual fluctuations in growth³¹. The repercussions on herbivore diversity and stability we observed can thus have important ecological consequences—for example with respect to predicting the probability of pest outbreaks following climate extreme events and subsequent tree growth variation³²—that would not be evident when ignoring such a temporal perspective. Understanding what promotes stability allows us to assess how ecosystems buffer disturbances, maintain trophic interactions, and preserve long-term biodiversity under changing environmental conditions.”

-The plantation/monoculture areas seem poorly integrated into the intro, results, and discussion. There are a few mentions of the monocultures and it seems like a missed opportunity to connect to human management meaningfully.

Response: Thanks for the useful comments! Our analyses are not so much monoculture vs. mixture comparisons but analyses along a tree species richness gradient. We have clarified this in the Introduction and in Fig. 1. Nevertheless, we agree that the implications of these analyses for management of monoculture plantations are important as well and we have added relevant text in the Introduction (see L94-L96) and Conclusions (see L354-L358).

In Introduction:

“Monoculture plantations represent a dominant forest management practice worldwide²², but their low biodiversity may reduce their capacity to buffer against environmental fluctuations and pest outbreaks.”

In Conclusions:

“Our findings suggest that monoculture plantations, despite their widespread use in forest management, may be more prone to herbivore instability due to low tree diversity. In contrast, mixed-species plantings appear to enhance stability, highlighting the value of biodiversity-oriented management for more resilient forest ecosystems.”

-I have a few questions about the “all” category for herbivore types, because on the figures points seem to be a subset (not including the specialists). Please explain/clarify.

Response: Thank you for this helpful comment. The category “all” refers to the entire herbivore community (i.e. generalists + specialists). Because generalist herbivores represent the majority of the individuals in the community, the patterns for “all” largely overlap with those of the generalists, resulting in similar ranges in the plots. This is therefore not a figure error, but a reflection of the community composition. We have clarified this in the figure legend (Fig. 2) to avoid misunderstandings.

-I also have several suggestions for minor changes for improved clarity (see in-line comments below).

In-line comments and section-specific recommendations:

Introduction:

General comments:

Overall, the intro would be strengthened by drawing more directly to diversity-stability theory (and acknowledging the debate/challenges/context dependency within). Because asynchrony is such an important part of the results, it also seems to need more discussion/description in the introduction.

In-line comments

Response: We have strengthened the connection to biodiversity–stability theory by explicitly mentioning some of its key mechanisms and context dependencies (L70-L77; L230-L240). We now also provide a more detailed introduction to the concept of species asynchrony and its potential role in stabilizing communities, thereby better preparing the reader for its prominence in our results (L100-L102):

In Introduction:

“Such stabilizing effects may operate through mechanisms such as averaging effects, negative covariance among species, and insurance effects⁸ and suggest that the current biodiversity crisis^{9,10} might have long-term negative consequences for ecosystem stability and ultimately human well-being^{6,11}. The full consequences, however, are difficult to assess. This is because, for one, biodiversity–stability relationships may vary with environmental (e.g. climate) conditions and across spatial and temporal scales^{7,12,13}, but we currently lack insights from many ecosystems.”

“In general, community stability can be partitioned into species asynchrony, i.e. a temporal misalignment of species dynamics, and the stability of species’ populations themselves¹⁴”

In Discussion:

“Moreover, our study highlights the added value of expanding biodiversity and ecosystem functioning relationships by explicitly considering stability relationships. While high average biomass production may reflect favorable conditions in some years, it may mask substantial interannual fluctuations in growth³³. The repercussions on herbivore diversity and stability we observed can thus have important ecological consequences—for example with respect to predicting the probability of pest outbreaks following climate extreme events and subsequent tree growth variation³⁴—that would not be evident when ignoring such a temporal perspective. Understanding what promotes stability allows us to assess how ecosystems buffer disturbances, maintain trophic interactions, and preserve long-term biodiversity under changing environmental conditions.”

110-111 This section leads the reader to believe they will be comparing among plant growth forms. I would clarify/modify to discuss tree traits more.

Response: Thank you for pointing this out. We have revised the text for clarity (see L117-L122):

“However, empirical insights on how diversity and stability are interlinked between herbivores and their host plants are available primarily for grasslands, which are characterized by a strong yearly biomass turnover of herbs and graminoids¹⁵. As long-lived organisms, trees are rather constant and predictable resources, and associated herbivores might thus be less affected by changes in the stability of host biomass production.”

134 Final hypothesis could be clarified.

Response: Further clarified (see L150-L152):

“3) community stability is primarily driven by asynchrony among dominant generalist herbivores, with specialist species being much rarer and their stability depending more strongly on population stability.”

184-196 I recognize this is a response to a reviewer, but it needs to be better connected to the hypotheses or better explained; as is it comes out of the blue and reads exactly like a response to a reviewer, rather than an integrated part of the study.

Response: Because we see these analyses more as sensitivity analyses. We realized that moving these additional analyses to the Supplementary Information is the best way to reduce confusion. Hence, we now present these analyses explicitly as sensitivity analyses (see Supplementary sections), pointing to it in the figure caption, while focusing the main text on the core results.

Results

General comments

I recognize that the authors are trying to strike the balance of enough methods details to give context but leave detailed methods below in the methods section; however, these results are presented relatively confusingly. Please take another readthrough and clarify and simplify. I suggest defining all abbreviations and explaining models more ecologically, rather than statistically (so that the reader does not need to know all the details of the models to understand the important differences). Start out more broad (maybe report overall direct and indirect effects) before getting into the details of each model. I would also suggest connecting the results directly to the hypotheses to better lead the reader through the figures/results. Changes to the topic sentences and/or organization would help.

Response: We have thoroughly revised the results to improve clarity and flow. Specifically, we now 1) define all abbreviations upon first use, 2) describe the models primarily from an ecological perspective in the caption of figure 1, 3) begin with a broader summary of overall direct and indirect effects before presenting model-specific details (see L160-L170), and 4) explicitly link each set of results back to the corresponding hypotheses (see L161, L171, L187). We also revised topic sentences and adjusted the organization of the text to guide the reader more clearly through the figures and results (see L155-L191).

In-line comments

142 Must use MOTU unabbreviated initially because it is the first mention here.

Response: Defined the abbreviation accordingly.

149 Same comment as above for MPD

Response: Defined the abbreviation accordingly.

Discussion

General comments:

Generally well written and interesting discussion, although I would have liked to see a higher proportion of the overall word count be in this section.

Response: We have expanded the discussion section by adding more synthesis of our key findings in the context of diversity–stability theory (see L216-L220), as well as elaborating on the implications for biodiversity conservation and ecosystem management (see L230-L242):

In-line comments

199 If the topic sentence specifies species richness of host trees, it seems slightly strange that the specifics in the next sentences are about functional diversity and tree growth.

Response: Rephrased to promote readability (L203-L207):

“Our study unravels how the temporal stability of abundance and species richness of herbivore communities are linked to changes in the species richness of their host tree communities. Specifically, the results show that indirect effects of tree species richness via tree functional diversity and tree growth on herbivore phylogenetic diversity...”

204 Yes, this is an important and ecologically intuitive result that should be highlighted

Response: We have now revised the text to highlight it more clearly (see L210-L220):

“ In particular, we found a clear and ecologically intuitive stabilizing effect of tree species richness on the temporal abundance and richness stability of specialist herbivores, whereas such effects were absent for generalists. For the overall herbivore community, path analysis indicated that this altogether resulted in less stable abundance dynamics at higher tree species richness via tree functional diversity, even though the direct relationship between tree species richness and herbivore stability was not significant in our linear models. These findings underscore the crucial role of host diversity in buffering specialist herbivores against environmental fluctuations and emphasizes that biodiversity–stability relationships can vary markedly across trophic groups depending on their level of dietary specialization.”

243-248 Same as comment above; needs to be rooted in ecology (or at least directly connected to your hypotheses) to make sense of why you replaced this.

Response: We have rephrased this section to better connect it to our hypotheses and to highlight the ecological rationale behind the variable choice in models (L263-L272):

“In the sensitivity analyses replacing herbivore phylogenetic diversity by herbivore abundance or richness (Figs S3, S4), the driving role of tree species richness behind these effects became even more evident. This was possibly due to the fact that these simpler metrics of biodiversity were not able to resolve the indirect pathways via tree asynchrony as accurately as a biodiversity metric accounting for phylogenetic (and functional) differentiation among herbivore species^{36,37}, underlining the importance of considering multiple metrics of biodiversity. Notwithstanding, the strong influence of tree species richness emphasizes the consequences that biodiversity loss at the host level can have for higher trophic-level diversity and, ultimately, stability.”

288 Same comment as above, integrate better into the intro to justify

Response: We now framed these additional questions as sensitive analyses and moved them to the Supplementary Information to reduce confusion (please see our responses to your above comment).

Methods

General comments

The study design (planted saplings across a range of species richnesses) should be mentioned in the intro even though the methods occur afterwards.

Response: We now briefly introduce the study design in the Introduction to provide readers with earlier context (see L130-L140):

“We used time series data (17 replicated sampling seasons spread across six years from 2017 to 2022) of a dominant group of insect herbivores (Lepidopteran larvae) and their host tree communities in a large-scale tree biodiversity experiment (BEF-China²⁷). The experiment was conducted at two sites located 4 km apart and our study included 52 randomly distributed plots reflecting a tree species richness gradient from monocultures to mixtures with up to 24 species. Specifically, we used linear models and path analysis to test the pathways that directly and indirectly (via tree and herbivore community asynchrony and population stability and via herbivore phylogenetic diversity) connect tree species richness and its effects on tree growth to the temporal stability of herbivore abundance and species richness (Fig. 1)”

In-line comments

325 How old are the saplings at the time of the experiment? If still saplings, I would make sure to say “tree saplings” early on (maybe in the intro justify why they are a particularly good system), in the results, and add at least a sentence in the discussion about how you may expect the results to change as the forest ages.

Response: Both experimental sites were established in 2009 and 2010, respectively, by planting young tree saplings (approximately one year old at the time of plantation). Thus, during the period of our study, the stands represent young experimental forests that were past the sapling phase and forming a closed canopy.

444 Is it “unrealistically” low stability? Isn’t high variability (and low stability) exactly what we would expect from rare species?

Response: We agree that the wording “unrealistically” was misleading. We have now rephrased this section to be more specific, clarifying that low stability is indeed expected for rare species. At the same time, we now treat this analysis as a sensitivity test and have moved the corresponding text to the Supplementary Information for clarity (see Supplementary section L16).

507-526 I assume this is a response to a previous reviewer, but to me the four models, rather than structuring a single model based directly on the interactions of interest in the hypotheses, feels disconnected from the main aims of the paper. Maybe having one sentence in the introduction that clarifies that you are specifically interested in the direct, indirect, microclimate, or both (or something that captures all four models) effects would help. This is also not clearly connected to the main results presented in the figures (rather than the supplementals).

Response: Thank you for pointing this out. We now provide a brief rationale in the Introduction for why we compared multiple model structures, and directly reference the overview Figure, which previously was in the supplementary material but is now integrated in Fig. 1. We also rephrased the model description in the methods to better connect these alternative models with our hypotheses (L552-564):

“First, we considered the initial model with all potential links included (Model 1). Second, we assumed that effects are entirely mediated by stability components, except for effects such as microclimate buffering that may remain in tree species richness. Therefore, we removed the direct connections between trees and

herbivore abundance and richness stability, except for effects of tree species richness (Model 2). Third, we assumed that there are no such effects of tree species richness, so we additionally excluded the direct pathways between tree species richness and herbivore stability metrics (Model 3). Fourth, we assumed that tree species richness only has indirect effects via tree stability components and functional composition. We hence removed the links between tree species richness and herbivore MPD/abundance/richness and population stability (Model 4). The same model structure selected for overall herbivores was applied to generalists and specialists to ensure comparability across groups.”

Figures

General comments

Figure 1 clearly sets up the “monoculture versus everything else”, which isn’t the way the analyses were run (more continuous). Consider which is the correct choice for the hypotheses and main messages of the paper.

Response: Now we slightly modified the figure 1 to represent low tree species plots rather than only monocultures.

Fig. 1 Conceptual figure and initial path model structure illustrating the determining mechanisms of herbivore community stability. Herbivore abundance asynchrony among species and community stability are coupled with the diversity and temporal growth rate stability of their host tree communities (Hypothesis 1), with herbivore community stability ultimately a destabilized by low tree species richness and b stabilized by high tree species richness (Hypothesis 2). Specifically, in **a** monocultures and species-poor mixtures, the pronounced fluctuations in tree growth may lead to reduced asynchrony among herbivore species, thereby destabilizing the herbivore community. Conversely, in **b** more diverse mixtures, complementary dynamics resulting from the asynchrony among tree species could contribute to the stabilization of plant communities. This stabilization effect extends to herbivore communities, promoting greater asynchrony among herbivore species and ultimately enhancing the stability of the herbivore community (Hypothesis 3). These relationships were investigated with **c** path models. Structure based on theoretical expectations and correlations among herbivore- and tree-based variables: mean phylogenetic distance (MPD), species asynchrony, population stability of herbivores; tree functional diversity (FD), species asynchrony, population stability of trees. Arrows indicate expected causal relationships. Blue lines are covariances retained in the path models. We assessed with four alternative models how direct vs. indirect effects of host tree-based metrics influence herbivore community dynamics. Model 1: both direct and indirect pathways from trees to herbivore stability; Model 2: restricting tree growth effects to indirect pathways via herbivore population stability and asynchrony, except for tree species richness; Model 3: assuming even species richness acts only indirectly; Model 4: assuming that tree diversity influences herbivores solely through effects on tree functional diversity, asynchrony, and population stability. Model 4 was selected for our final analyses.

Figure 2d,e,f. If dark red is “all”, why do the lines and data only span the range of the generalist species? Is this just a figure error or does this represent the analyses? Needs a better explanation of “all” either way.

Response: The category “all” refers to the entire herbivore community (i.e. generalists + specialists). Please also see our response to you comment above.

“All’ herbivores includes the entire community (generalists and specialists) analyzed together...”

Figure 4 write out functional diversity on the figure legend

Response: Done.

Fig. 4 Effects of tree species richness on herbivore richness stability. Bars show summed effects of tree species richness on the abundance and richness stability of all, generalist and specialist herbivores, respectively. Effect sizes were calculated by summing indirect effects of tree species richness via tree functional diversity, tree asynchrony, tree population stability, herbivore phylogenetic diversity, herbivore asynchrony, herbivore population stability, and herbivore abundance stability. The different colors show effects of tree species on herbivore stability via tree functional diversity, tree asynchrony, and tree population stability, respectively. Effect sizes were calculated as the product of standardized path coefficients connecting each predictor with herbivore components, summed over the individual predictors of each component for positive and negative effects on herbivore stability metrics, respectively. Black T-shaped lines indicate the total effects of tree species richness on herbivore stability metrics. Note that tree species richness, population stability, abundance stability and richness stability of herbivores were log-transformed. Stability measures are based on the inverse of the coefficient of variation (eqn. 3). See Supplementary Results and Fig. S7 for effects including non-significant pathways.

Supplemental Figures

General comments

Fig S1. Say which was the final model structure used.

Response: Clarified accordingly, now in Fig.1

“Model 4 was selected for our final analyses.”

Fig S2. Tree species richness does not appear in this figure, does it? Check to make sure these are the right figures or captions. Also same question as in figure 2: why does “all” not encompass the specialist species?

Response: We now removed tree species richness from the caption. Please see our responses above regarding the “all” herbivores

Fig S3. I would change the title of this to encompass the replacement directly rather than in parentheses.

Response: Changed as suggested.

Fig S4. Same comment as S3

Response: Changed as suggested.

Fig S5 Same comment as Fig 2, and aren't some of these bivariate relationships in Fig 2? (e.g., e?, which also has no label)?

Response: The Figure depicts results for the alternative stability metric. Added the label for e.

Fig S6 Be clear about ho these are different from Fig 3.

Response: We clarified it further in the figure legend to easily distinguish it from Fig. 3.

Asynchrony and functional diversity couple herbivore community dynamics to host plant diversity

Ming-Qiang Wang, Georg Albert, Carlo L. Seifert, Douglas Chesters, Helge Bruelheide, Jing-Ting Chen, Andréa Davrinche, Sylvia Haider, Shan Li, Yi Li, Goddert von Oheimb, Tobias Proß, Keping Ma, Xiaojuan Liu, Arong Luo, Andreas Schuldt*, Chao-Dong Zhu*

RESPONSE TO REVIEWER 1

Reviewer #1 (Remarks to the Author):

Thank you for this new, revised version. My comments have all been addressed satisfactorily. I have on final remark: In response to a reviewer comment, the conclusions were extended. In my opinion, the conclusions included in lines 353-358 do not reflect the study results, as the destabilization of herbivore communities due to biodiversity loss only holds for specialized herbivores. I suggest to include a little more differentiation here.

Response: Thank you for this insightful remark. We agree that the destabilization of herbivore communities due to biodiversity loss primarily applies to specialist herbivores. We have now revised the conclusion to specify that monocultures are especially vulnerable to instability driven by specialist herbivores, while mixed-species plantings can mitigate such risks by diluting host availability.

“Our findings suggest that monoculture plantations, despite their widespread use in forest management²², may be more prone to herbivore instability driven by specialist herbivores that can rapidly build up on their preferred hosts under low tree diversity. In contrast, mixed-species plantings dilute host availability and thereby stabilize herbivore communities, highlighting the value of biodiversity-oriented management for more resilient forest ecosystems.”

RESPONSE TO REVIEWER 4

Reviewer #4 (Remarks to the Author):

Overall statement:

The authors have sufficiently addressed my previous concerns and the other reviewers'; the manuscript will make a very nice contribution to the field, and I once again commend the authors on their work. Below are a few minor suggestions for small clarifications.

Response: We appreciate the careful reading and have addressed all minor suggestions accordingly to further improve the clarity and precision of the manuscript.

In-line comments:

76 and we lack, rather than but we lack

Response: Changed accordingly.

141-152 May want to consider specifying a, b, c if possible for Figure 1 rather than refer generally to Figure 1 multiple times in these sentences

Response: We have now specified the corresponding panels when referring to Figure 1 to improve clarity and readability.

236 the repercussions of what? Clarify this sentence/few sentences

Response: We have specified this point more clearly by indicating that it refers to “the repercussions of tree diversity”.